# Targeted systematic evolution of an RNA platform neutralizing DNMT1 function and controlling DNA methylation

Carla L. Esposito [1] ✉, Ida Autiero [2,15,16], Annamaria Sandomenico [3,16], H. Li [4], Mahmoud A. Bassal [5,6], Maria L. Ibba [1], Dongfang Wang[7], Lucrezia Rinaldi[8], Simone Ummarino[6,8], Giulia Gaggi[8], Marta Borchiellini[9,10], Piotr Swiderski[11], Menotti Ruvo [3,12], Silvia Catuogno[1], Alexander K. Ebralidze[6,8], Marcin Kortylewski [7] ✉, Vittorio de Franciscis [1,13] ✉ & Annalisa Di Ruscio [8,9,14] ✉

DNA methylation is a fundamental epigenetic modification regulating gene expression. Aberrant DNA methylation is the most common molecular lesion in cancer cells. However, medical intervention has been limited to the use of broadly acting, small molecule-based demethylating drugs with significant side-effects and toxicities. To allow for targeted DNA demethylation, we integrated two nucleic acid-based approaches: DNMT1 interacting RNA (DiR) and RNA aptamer strategy. By combining the RNA inherent capabilities of inhibiting DNMT1 with an aptamer platform, we generated a first-in-class DNMT1-targeted approach – aptaDiR. Molecular modelling of RNA-DNMT1 complexes coupled with biochemical and cellular assays enabled the identification and characterization of aptaDiR. This RNA bio-drug is able to block DNA methylation, impair cancer cell viability and inhibit tumour growth in vivo. Collectively, we present an innovative RNA-based approach to modulate DNMT1 activity in cancer or diseases characterized by aberrant DNA methylation and suggest the first alternative strategy to overcome the limitations of currently approved non-specific hypomethylating protocols, which will greatly improve clinical intervention on DNA methylation.

DNA methylation is a key epigenetic signature implicated in the regulation of gene expression[1–3]. Methylation of CpG-rich (mCpG) promoters is carried out by members of the DNA methyltransferase (DNMT) family (DNMT1, DNMT2, DNMT3A, DNMT3B and DNMT3L)[3–5]. Several studies have established a link between aberrant promoter DNA methylation and cancer. Reduced DNA methylation contributes to genomic instability while site-specific aberrant methylation of gene promoters, namely for tumour suppressor genes and transcription factors, results in specific gene silencing[5,6]. Abnormal epigenetic modifications likely occur in the early stages of the tumour development and are reversible thereby offering the possibility to restore normal gene function abrogated by malignant cellular transformation. In light of this, "epigenetic targeting" brought about by specific and effective inhibitors could lead to clinically relevant strategies for cancer therapy[7].

Thus far, two nucleoside-based compounds, 5-azacytidine (5-Aza) and 5-aza 2′-deoxy-cytidine have been approved as global DNA demethylating agents. However, the lack of selectivity, the toxicity and the chemical instability have raised serious concerns for the use of these nucleoside analogues[1,8–10], revealing the clinical need to develop smarter and safer epigenetic therapeutic options.

A full list of affiliations appears at the end of the paper. ✉e-mail: c.esposito@ieos.cnr.it; MKortylewski@coh.org; vittorio.defranciscis@irgb.cnr.it; adirusci@bidmc.harvard.edu

Nucleic-acid aptamers are a promising class of three-dimensional structured oligonucleotides that serve as high-affinity ligands and potential antagonists of disease-associated proteins[11]. They are usually selected from a large random sequence library through a combinatorial chemistry strategy named SELEX (Systematic Evolution of Ligands by EXponential enrichment)[12,13]. Besides being cost-effective and relatively easy to manipulate, the small size of aptamers (6–30 kDa) enables them to access binding pockets physically inaccessible to macromolecules. In addition, they are not immunogenic and exhibit low toxicity, while retaining high affinity for their binding targets. These features make aptamers a valuable tool in clinical diagnosis, therapy, and targeted delivery[13,14].

Previously, we discovered a class of RNAs able to inhibit DNMT1 enzymatic activity and regulate DNA methylation[15]. Our findings showed that a non-coding RNA, named *extra coding-CCAAT-enhancer-binding proteins alpha* (*ecCEBPA*), originating within *CEBPA* locus could interact with and inhibit DNMT1, thereby preventing methylation of *CEBPA* locus. Globally, we proved that the DNMT1 site-specific sequestration by DNMT1-interacting RNAs (DiRs) occurs at multiple loci and is dependent upon the presence of RNA stem-loop-like-secondary structures. Further, we identified the minimal 22 nucleotide-long recognition motif required to block DNMT1 activity[15], delineating an RNA-mediated mechanism controlling DNA methylation establishment.

Herein, we adopt a "doped" SELEX approach to generate DNMT1-neutralizing RNA aptamers with enhanced affinity, specificity and stability than the natural existing counterparts. Through a combination of in silico modelling, in vitro biochemical, cellular-based assays and in vivo validation, we demonstrate the applicability of this targeted DNMT1-specific platform—aptaDiR, as the first of its kind RNA-based approach to correct aberrant DNA methylation that holds promise for

the treatment of cancer or other diseases characterized by aberrant DNA methylation.

## Results

### Evolution of anti-DNMT1-specific RNA aptamers−aptaDiRs

In our previous study, we identified a new mechanism for RNA-dependent regulation by DNMT1-interacting RNA (DiR)[15]. The DiR function relied on the presence of a 22-nucleotide (nt)-long stem-loop-like RNA structure (R5), that is sufficient to bind the DNMT1 catalytic domain and inhibit its activity[15]. Since specific tridimensional structures are involved in the formation of the RNA-DNMT1 complex[15], we sought to model in silico the interaction between R5 or its mutant R5 (mutR5, which is unable to fold in stem-loop-like-secondary structures), with the human (h)DNMT1 catalytic region. The coordinates of the murine DNMT1 solved structure (PDB: 4d4a) in complex with its DNA substrate were used as a template (see methods for details). This analysis revealed a different arrangement of R5 with respect to the mutR5. Indeed, while the former adopts a stem-loop-like architecture that perfectly accommodates within the hDNMT1 catalytic region, thereby mimicking and competing out its natural DNA substrate, the latter adopts a conformation that hampers an adequate fitting within the DNMT1 binding site (Fig. 1a).

To evolve anti-DNMT1-RNA aptamers (hereafter also referred to as aptaDiRs) with increased affinity and stability, we built up a "doped" protein-SELEX strategy by using as a starting pool R5 variants with three fully randomized short regions (Fig. 1b, c and "Methods"). The addition of 2′-Fluoro-Pyrimidines (2′F-Py) analogues at each position provided enhanced resistance to degradation of the RNA library. The 2′F-Py modified R5 (herein referred to as DNMT1 bait) preserves its binding to DNMT1 as assessed by direct-Enzyme-Linked Oligonucleotides Assay

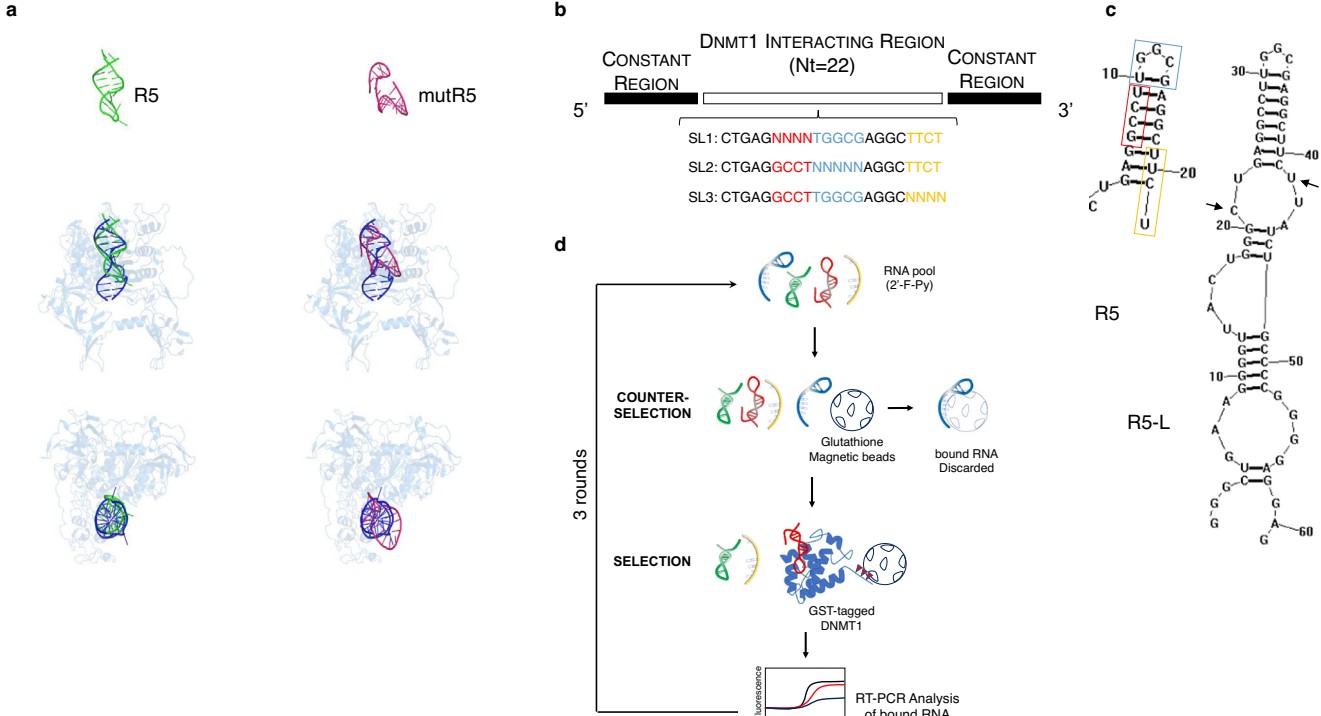

**Fig. 1 | R5 models and aptamer selection process. a** Cartoon representation of the tridimensional models of R5 (green) and mutR5 (magenta) and their complexes with the DNMT1 (blue) protein. The DNA substrate of the X-ray structure used guide to build DNMT1 complexes is also shown in blue. **b** Diagram of the site-specific randomized sub-libraries (SL1, SL2 and SL3) used as starting pool for the SELEX cycles. **c** Stem-loop predicted structures of R5 and long R5 (R5-L). The black arrows indicate R5 sequence within R5-L. Regions within R5 that were randomized for the SELEX starting pool are boxed (SL1: red; SL2: light blue; SL3: yellow). **d** Scheme of the SELEX rounds. Each round includes steps of: i) incubation of the RNA pool with glutathione magnetic beads (counter-selection); ii) recovering of unbound sequences; iii) incubation of the unbound sequences with purified GST-tagged DNMT1 protein (selection); iv) partitioning of the bound sequences with glutathione coupled magnetic beads; v) recovering and amplification of bound sequences by RT-PCR.

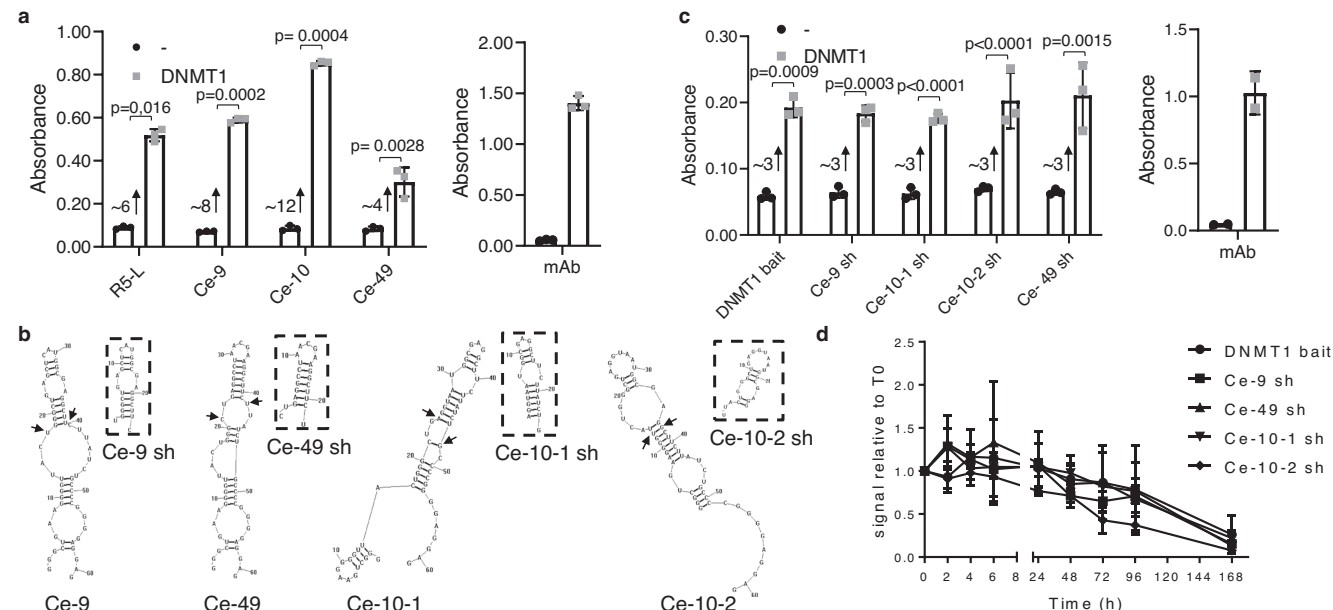

**Fig. 2 | Analyses and optimization of individual aptamers from SELEX. a** Binding of biotinylated selected aptamers from SELEX and R5-L on DNMT1 purified protein was analyzed by ELONA (*left panel*). Anti-DNMT1 antibody (mAb, *right panel*) was used as a positive control. Error bars depict mean ± SD. Experiment is representative of three independent biological replicates. Statistics by two-tailed *t*-test is indicated (*n* = 3 biologically independent samples). Fold increase of binding is reported. **b** Predicted secondary structures of Ce-49, Ce-9 and Ce-10 and designed shortened aptamers (boxed). Two different folding and correspondent short aptamers were predicted for Ce-10 and indicated as Ce-10-1 and Ce-10-2, respectively. The black arrows indicate the sequences of the short versions within the corresponding long aptamer. **c** Binding of 3′- biotinylated short aptamers and DNMT1 bait on DNMT1 purified protein was analyzed by ELONA. Error bars depict mean ± SD. Experiment is representative of three independent biological replicates. Statistics by two-tailed *t*-test is indicated (*n* = 3 biologically independent samples). Anti-DNMT1 antibody (mAb, *right*) was used as a positive control. Fold increase of binding is reported. **d** Short aptamers and DNMT1 bait serum stability were measured in 85% human serum for indicated times. At each time point, RNA-serum samples were collected and evaluated by electrophoresis with 15% denaturing polyacrylamide gel. The intensity of the bands was quantified and expressed relative to T0. Means ± SD with the corresponding data points are shown. **a**–**d** Source data are provided as a Source Data file.

(ELONA) and shows a binding efficiency comparable to the unmodified R5 (Supplementary Fig. 1a).

The SELEX library was designed starting from a construct (R5-L) containing variants of *ecCEBPA* R5 as the DNMT1 targeting region flanked by two constant 20- and 19-nt long naïve regions, at the 5′ and 3′ ends, respectively. The T7 promoter sequence engineered upstream of the 5′ end allowed transcription and amplification of the RNA construct (Fig. 1b). Importantly, the R5-L retained the stem-and-loop-like structure required for the DNMT1–RNA interaction (Fig. 1c). Unlike standard protein-SELEX protocols starting from a high complexity library ($10^{14}$ to $10^{15}$ variable sequences), a "variation" of R5 was designed to introduce stringent constraints on the possible folding of the selected aptamers. This type of stringency was obtained by mixing three different starting libraries wherein only a short part of the DNMT1 bait sequence was fully randomized, while the remaining portion was not. Specifically, three distinct libraries (SL1, SL2 and SL3) were produced by degenerating three different regions (4–5 bases each) from the 5′, the central loop and the 3′ of the 22-nt DNMT1 bait (Fig. 1b, c). The three libraries were mixed at an equimolar concentration and used as template for the transcription of the 2′F-Py modified RNA starting pool.

This strategy enabled us to scale down library complexity by nearly $10^{12}$-fold as compared to a standard SELEX. In this instance, fewer cycles were predicted to be necessary for the enrichment[16], and only 3 rounds of SELEX were performed using the purified Glutathione S-Transferase (GST)-tagged DNMT1 protein as target. As schematized in Fig. 1d, at each round, the pool was first incubated with glutathione coupled magnetic beads for a counter-selection step. The bound sequences were separated using a magnetic separator and the unbound aptamers were used for the selection on GST-tagged DNMT1. Following the selection step, bound sequences were partitioned on glutathione coupled magnetic beads and recovered by reverse transcription PCR (RT-PCR). During the three rounds, we used an increasing number of final washes to progressively enhance the stringency of the selection and to recover aptamers with higher affinity for the target (Supplementary Table 1). Consistently, a strong binding performance of the enriched library after 3 rounds of selection was confirmed by ELONA (Supplementary Fig. 1b).

### Selection of aptaDiRs
After the third SELEX round, the final enriched pool was cloned using TA cloning system and approximately seventy clones isolated and sequenced. The sequences were aligned by *Muscle* algorithm[17] and clustered into families. Among the analyzed clones (Supplementary Fig. 2), variants coming from the three SLs were equally represented. The Ce-9 and Ce-10 aptamers, identified as the most enriched in the 3 clusters of sequences (Supplementary Fig. 3a), exhibited two mismatches between each other and three as compared to the parental R5 (Supplementary Fig. 3b) while the third aptamer Ce-49, chosen from a separate cluster, displayed full disruption of the seeding sequence with respect to the original counterpart (Supplementary Fig. 3b).

AptaDiRs binding was confirmed by ELONA. A binding efficiency comparable or even stronger than R5-L was detected for all analyzed sequences in a range of 200 nM (Fig. 2a). Efficient and cost-effective chemical synthesis is a key aspect of aptamer optimization. Thus, the initially identified 61-nt long aptaDiR sequences was reduced to an ideal 22/25-nt length, without altering the stem-loop-like structures critical for the interaction with DNMT1. The shortened (sh) aptamers: one for Ce-9 or Ce-49 (indicated as Ce-9 sh and Ce-49 sh, respectively) and two for Ce-10 (Ce-10-1 sh, Ce-10-2 sh), displayed the same hairpin-loop predicted structure as the longer sequences (Fig. 2b) and preserved the binding ability to DNMT1 (Fig. 2c), thus confirming

the stem-loop-like structures to be sufficient for the aptamer-DNMT1 interaction.

We then analyzed the stability of DNMT1 bait vis-à-vis the selected aptamers when incubated with high concentrations of human serum (85%) at 37 °C, over an extended time. The four aptamers and the DNMT1 bait remain almost undigested up to 48 h, likely owing to the presence of the 2′F-Py modification (Fig. 2d and Supplementary Fig. 4).

In summary, our approach led to the identification of four very stable short RNA aptamer ligands for DNMT1.

## Characterization of binding and target specificity of the aptaDiRs

To probe the biomolecular interactions and test by direct binding the aptamer affinity to DNMT1, we initially took advantage of the Bio-Layer Interferometry technology[18]. We tested the four aptaDiRs and found that the best dose-dependent binding to DNMT1 was detected for Ce-49 sh and Ce-10-2 sh (Supplementary Fig. 5a, b). The affinity measurement (Fig. 3a, b) provided $K_D$ values of $79.2 \pm 23.7 \times 10^{-9}$ M ($R2 = 0.9646$) and $66.4 \pm 24.0 \times 10^{-9}$ M ($R2 = 0.9426$) for Ce-49 sh and Ce-10-2 sh, respectively. Dose–response binding analyses were also performed with the DNMT1 bait at concentrations ranging between 100 nM and 2 μM, since no binding was detected at lower concentrations (Supplementary Fig. 5c). Data fitting (Fig. 3c) provided for this molecule a $K_D$ value of $0.6 \pm 0.1 \times 10^{-6}$ M ($R2 = 0.9017$), about 10 times higher as compared to that of the aptaDiRs. The binding curves also showed stronger signals about 1.25 and 1.80 nm (at the highest concentration) for Ce-49 sh and Ce-10-2 sh, respectively, as compared to DNMT1 bait (about 0.17 nm). This difference further affirms the increased affinity of the two generated aptaDiRs (Fig. 3a–c). In accordance with previous findings[15], no binding was detected for the mutR5 sequence which is unable to fold in stem-loop-like structures and was used as a negative control (Supplementary Fig. 5d). Therefore, Ce-49 sh and Ce-10-2 sh were chosen for additional investigations. Firstly, we confirmed binding affinity to DNMT1 using microscale thermophoresis (MST) (Fig. 3d–f and Supplementary Figs. 6–9). In Fig. 3d–f, the binding curves plot aptaDiRs' or DNMT1 bait's bound fraction as a function of the concentration. The aptaDiRs show dose-dependent and saturable curves which reflect $K_D$S of $119 \pm 34 \times 10^{-9}$ M (Ce-49 sh) and $106 \pm 42 \times 10^{-9}$ M (Ce-10-2 sh). As expected, DNMT1 bait displays a higher $K_D$ value ($0.44 \times 10^{-6}$ M) as compared to the aptaDiRs and the negative control mutR5 does not show any binding within the concentration range explored (Supplementary Fig. 8). Of note, the seemingly higher $K_D$S, as determined by MST, were in the same range of magnitude of those obtained through BLI (79 nM and 66 nM), suggesting consistency of the two measurements. Technical differences between the two approaches could indeed account for the marginal change observed.

The improved binding ability of the selected aptaDiRs was also validated by RNA Electrophoretic Mobility-Shift Assays (REMSA) (Fig. 3g). The ability to bind the native target protein in the proper cellular context is fundamental for the in vivo application of aptamers selected with recombinant proteins. Therefore, to check aptaDiRs' binding to the endogenous DNMT1 protein, we carried out an aptaDiR-mediated pulldown assay. Extracts from K562 leukaemia cell model, showing high levels of DNMT1, were incubated with biotin-tagged DNMT1 bait, Ce-49 sh or Ce-10-2 sh and the complexes were purified on streptavidin-coated beads, followed by immunoblotting with anti-DNMT1 antibody. All tested aptaDiRs display interaction with the native DNMT1 (Fig. 3h).

Next, we evaluated selectivity and specificity characteristics of Ce-49 sh and Ce-10-2 sh for DNMT1 as compared to the DNMT1 bait, by performing comparative binding experiments with the main members of the DNMT family proteins: DNMT3A and DNMT3B (Fig. 4a, b). As for DNMT1 bait (Fig. 4c), no significant interaction was recorded between the tested aptaDiRs and the two control proteins confirming the high

specificity of aptaDiRs against DNMT1. This was further proven by the absence of binding to the unrelated chromatin modifier lysine acetyltransferase 5 (KAT5) protein (Fig. 4c).

Thereafter, we assessed the aptaDiRs' binding to the human serum albumin (HSA) the most abundant plasma protein. HSA binds nucleic acids in a non-specific manner through its positive charges and may reduce the bioavailability of circulating aptamers, thereby limiting their clinical applicability. Remarkably, no interaction was detected by incubating Ce-49 sh and Ce-10-2 sh at concentrations up to 750 nM with HSA, (Fig. 4d and Supplementary Fig. 10).

Taken together, these data show that Ce-49 sh and Ce-10-2 sh are high affinity and specific ligands for DNMT1 with apparent in vitro $K_D$S within the nanomolar range, appealing features to proceed to clinical studies.

## aptaDiR functional characterization

To examine the ability of the selected aptaDiRs (Ce-49 sh and Ce-10-2 sh) to inhibit DNMT1-mediated methylation, we performed an in vitro DNMT1 inhibition assay. The enzymatic activity of DNMT1 purified protein was measured in the absence or presence of DNMT1 bait or the aptaDiRs. All the sequences blocked DNMT1 as shown by the 50% reduction of its activity (Fig. 5a). Consistently, a reduction of 40 to 60% DNMT1 activity was reached when testing nuclear extracts obtained from K562 cells previously transfected with the Ce-49 sh and Ce-10-2 sh, as compared to the untreated cells (Fig. 5b). As expected, no effect was observed using extracts from cells transfected with mutR5 (indicated as Cont.). Further, the DNMT1 bait, Ce-49 sh and Ce-10-2 sh were unable to interfere in vitro with the enzymatic activity of DNMT3A/B (Fig. 5c), thus confirming the high affinity and selectivity of the aptaDiRs for DNMT1.

Finally, we evaluated the transcriptional activation brought about by aptaDiRs in K562 wherein the *CEBPA* promoter is methylated and the mRNA is expressed at low-to-undetectable levels[15]. Upon transfection with the aptaDiRs, we observed effective increase of *CEBPA* levels (Fig. 5d), whereas no changes occurred when the control oligonucleotide was used. As a result of DNMT1 inhibition and *CEBPA* upregulation[19,20], K562 cell viability was reduced 60 to 50% upon aptamer transfection (Fig. 5e) when compared to the non-targeting control.

To estimate the extent of DNA methylation resulting from the aptaDiRs, genome-scale methylation profile was assessed using the EPIC array platform on K562 cells transfected with Ce-49 sh and Ce-10-2 sh. The differential methylation analyses revealed significant reduction of DNA methylation across thousands of CpG covered by the array in the aptaDiR-treated cells as compared to the control (Fig. 6a). Nearly 16,000 and 14,000 differentially methylated regions (DMRs) were detected for the Ce-49 sh and Ce-10-2 sh samples, respectively (Fig. 6b, c) with more than 60% overlap between the two treatment groups (Fig. 6c), likely resulting from a distinct mode of action of the two aptamers based on different molecular stabilities of the aptaDiR–DNMT1 complexes as described below. Interestingly, gene ontology (GO) analyses ("biological process") of genes corresponding to the overlapping hypomethylated CpGs (Supplementary Data 1) included among the top ranked, GO terms belonging to epigenetic modification, regulation of transcription and gene expression, consistently with the function of the aptaDiRs (Fig. 6d). Changes following aptaDiRs treatment within the *CEBPA* locus was assayed by pyrosequencing, owing to the paucity of the probes covering the *CEBPA* locus in the EPIC array. Consistent and significant demethylation was observed at the *CEBPA* proximal promoter and coding region for Ce-49 sh and at the proximal promoter for Ce-10-2 sh (Fig. 6e). Remarkably, only Ce-49 sh did not show any effect on the DNA methylation profile of LINE-1 and compared to 5-Aza treatment (Supplementary Fig. 11) suggesting that a safer and highly selective therapeutic profile may be achieved by the use of Ce-49 sh aptaDiR.

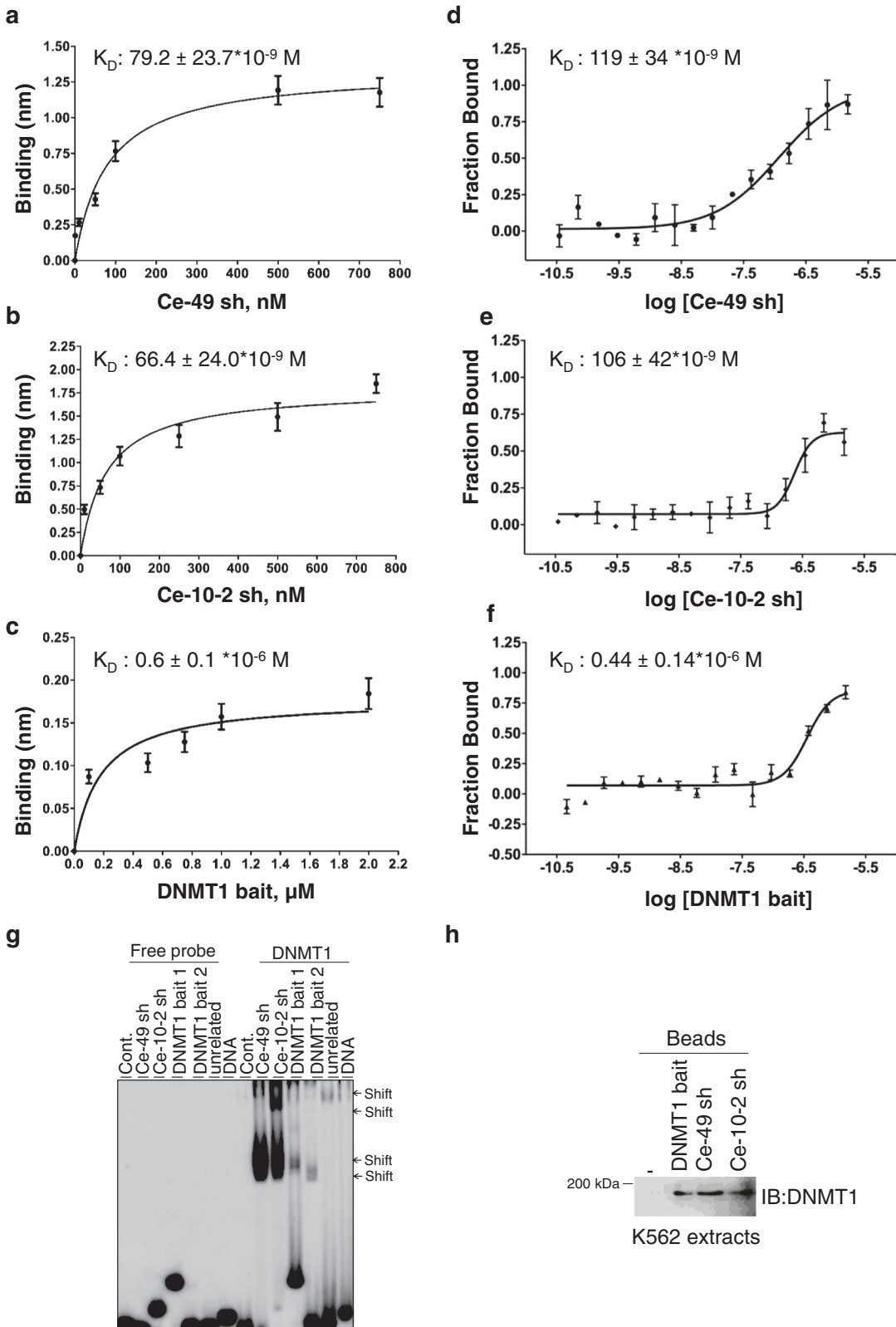

**Fig. 3 | AptaDiR affinity and in vivo binding to DNMT1.** Binding curves derived from BLI analyses of Ce-49 sh (**a**), Ce-10-2 sh (**b**) or DNMT1 bait (**c**). Curves were fitted with a 1:1 binding model using GraphPad Prism 6. Individual values and the errors calculated by fitting binding curves using Blitz Pro 1.2 software are shown. **d**–**f** MST dose–response curves for the interaction analysis of aptaDiRs or DNMT1 bait reported as Fraction Bound. **d** Ce-49 sh; **e** Ce-10-2 sh; **f** DNMT1 bait. Means values and standard deviation of independent biological replicates ($n = 3$ in d and e;

$n = 2$ in f) are shown. **g** The interaction of DNMT1 and DNMT1 bait or indicated aptamers was analyzed by EMSA. DNMT1 bait 1 and 2 refer to different folding. **h** AptaDiR-mediated pulldown of DNMT1. Protein extract from K562 cells were incubated with the biotinylated DNMT1 bait, Ce-49 sh and Ce-10-2 sh. Bound proteins were purified on streptavidin beads and immunoblotted with anti-DNMT1 antibodies. Experiment was repeated independently two times with similar results. **a**–**h** Source data are provided as a Source Data file.

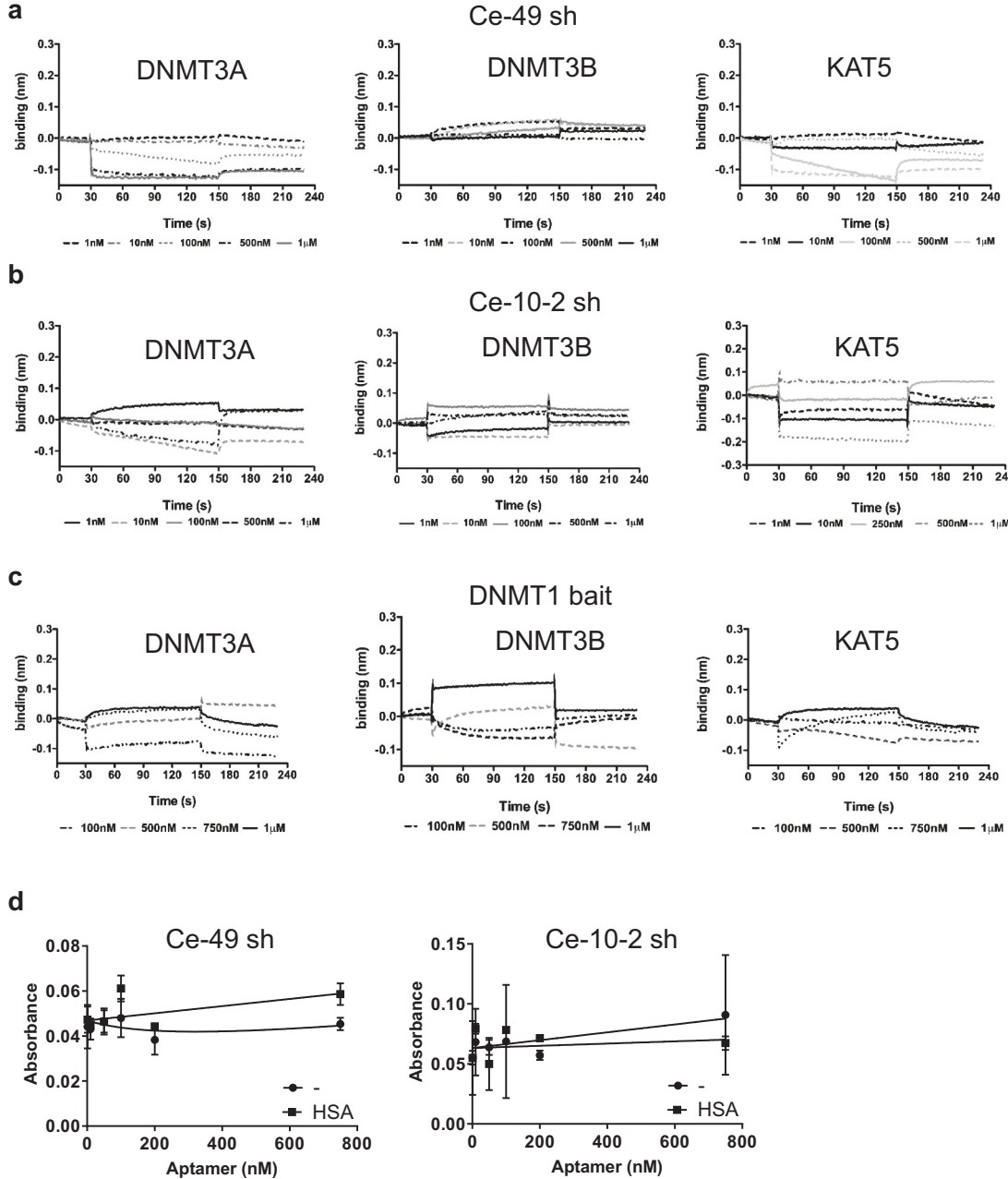

**Fig. 4 | AptaDiR specificity.** Binding measured by bio-layer interferometry of Ce-49 sh (**a**), Ce-10-2 sh (**b**) and DNMT1 bait (**c**) to DNMT3A (*left panels*), DNMT3B (*middle panels*) and KAT5 protein (*right panels*) immobilized on separate biosensors. Aptamers were tested at the reported concentrations. **d** ELONA assay of indicated biotinylated aptamers on plates left uncoated (-) or coated with HSA. Bars depict mean ± SD. Experiment is representative of two independent biological replicates. Source data are provided as a Source Data file.

Altogether, our results support the potential use of these newly generated molecules as an approach to block DNMT1 activity and reduce cell viability of cancer cells.

**Molecular dynamic simulations of DNMT1-aptaDiR complexes**
To obtain important insights on how aptaDiR binding takes place, we conducted in silico molecular simulations and dissected the structural and dynamic properties of DNMT1 interaction with the selected aptamers. Simulations were performed in explicit waters for 300 nanoseconds (ns) with the parental R5–, Ce-49 sh– or Ce-10-2 sh–DNMT1 complexes, respectively (Fig. 7a). The values of the root mean square deviation (RMSD) (Supplementary Fig. 12) of the trajectory structures versus the starting models indicate stability of the system during the entire simulation with slightly dissimilar

behaviour of the Ce-10-2 sh–DNMT1. In Ce-10-2 sh–DNMT1, the RMSD values exhibited by either the complex or each individual component were higher to some degree than those observed in R5-DNMT1 and Ce-49 sh–DNMT1 complexes. Additionally, the number of persistent hydrogen bonds within the last 150 ns simulation (Supplementary Figs. 13 and 14) resulted more stable at the aptamer-protein interfaces for the R5 and Ce-49 sh complexes than the Ce-10-2 sh-DNMT1 interface. Ce-10-2 sh shows a slightly different binding mode in targeting DNMT1 compared to the aptamers R5 and Ce-49 sh with lower hydration level at the interface with the protein (Supplementary Fig. 15). To further confirm these points, we performed three additional simulations of 300 ns for each complex by applying a new force field properly designed for RNA molecules (see methods for details). The percentage of hydrogen bonds

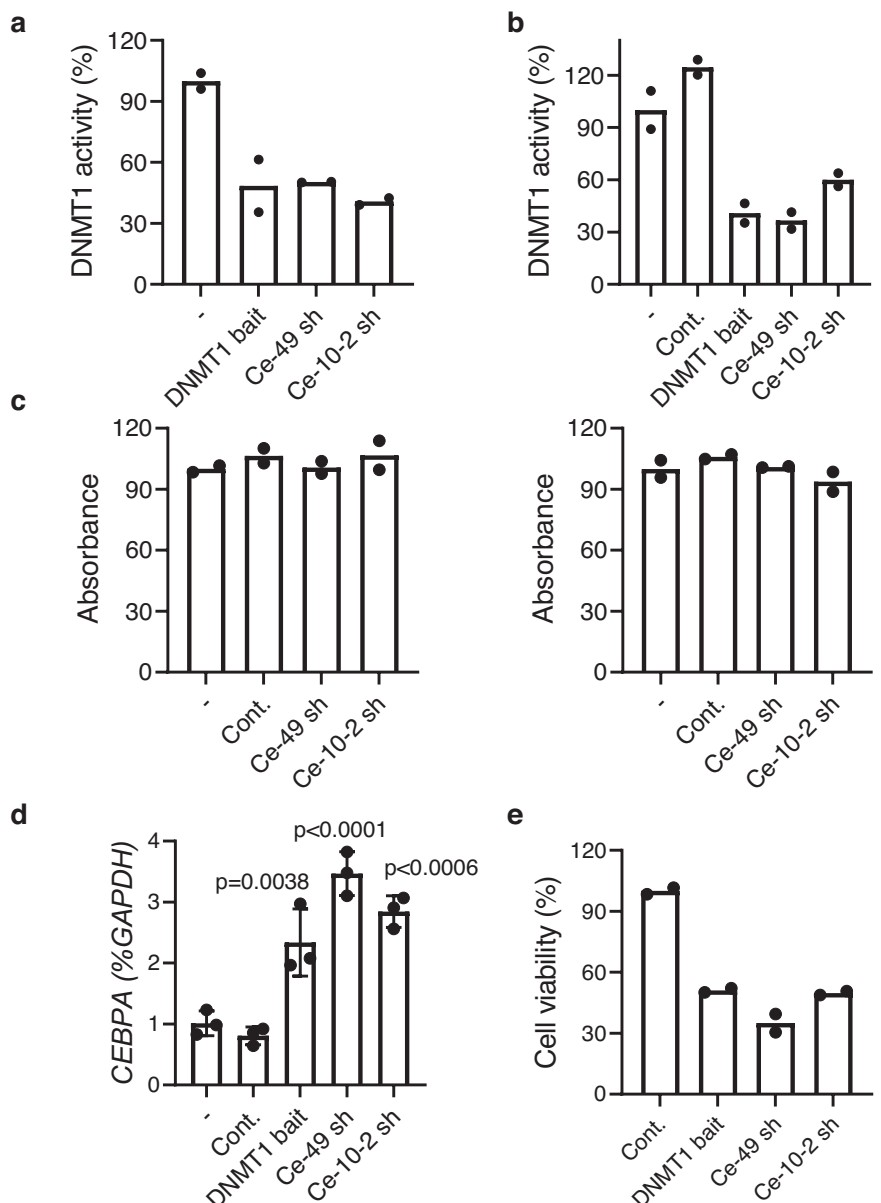

**Fig. 5 | Functional activity of aptaDiRs. a** Activity of purified DNMT1 protein was analyzed in vitro by DNMT1 inhibitor screening assay in the absence (-) or in the presence of indicated aptaDiRs or DNMT1 bait. Results are expressed as percentage relative to the activity of DNMT1 protein alone. **b** DNMT1 inhibitor screening assay was performed with nuclear extracts from K562 cells left untreated (-) or transfected with DNMT1 bait, indicated aptaDiRs or mutR5 used as a control (Cont.). Results are expressed as percentage relative to the activity detected with untreated sample. **c** Activity of purified DNMT3A (*left*) or B (*right*) proteins was analyzed in vitro in the absence (-) or in the presence of indicated aptaDiRs and expressed as percentage relative to the activity of DNMT protein alone. In (**a**–**c**) Mean values and

the corresponding data points are reported. Experiments are representative of two independent biological replicates with similar results (*n* = 2). **d** Levels of *CEBPA* were analyzed by RT-qPCR in K562 cells left untreated (-) or transfected with indicated aptamers or Cont. for 72 h. Mean ± SD with the corresponding data points is reported. Experiment is representative of three independent biological replicates. Statistic by one-way ANOVA (*versus* Cont.) is indicated (*n* = 3 biologically independent samples). **e** Cell viability of K562 cells transfected with indicated aptamers or Cont. for 72 h. Means and the corresponding data points are reported. Experiment is representative of two independent biological replicates (*n* = 2). **a**–**e** Source data are provided as a Source Data file.

computed considering the overall frames derived from the last 150 ns of each additional run, showed that R5 and Ce-49 sh complexes retain a high percentage of significant contacts with the DNMT1, keeping the stem-loop global arrangement till the end of all their simulations (Fig. 7b). DNMT1 interacted with R5 and Ce-49 sh harbouring both the 3' and 5' arms of the aptamers in a comparable fashion, connecting to both these aptamers via Arg1493, Arg1509, Arg1241, Arg1313 and Thr1528 residues. Instead, Ce-10-2 sh interacted with Arg1493, Arg1315 and Lys984 of the protein, but with low preservation of these contacts and showing a lower stability profile (Fig. 7b, Supplementary Fig. 16 and Supplementary Table 2). The hydration

level or ionic contribution at the aptamer-protein interfaces were comparable among the additional simulations (Supplementary Figs. 14 and 17). The differences of Ce-10-2 sh with respect to R5 and Ce-49 sh aptaDiR, although not so evident as previously observed, still suggest a higher variability of the molecular recognition between Ce-10-2 sh and the protein, that disfavours the binding preservation and the role of the solvent. The RMSD and hydrogen bonds profiles corroborate the observations emerged by the previous trajectories (Supplementary Fig. 18). The representative structures extracted either from all the runs, consistently demonstrated that R5 and Ce-49 sh are able to similarly accommodate within the binding region of

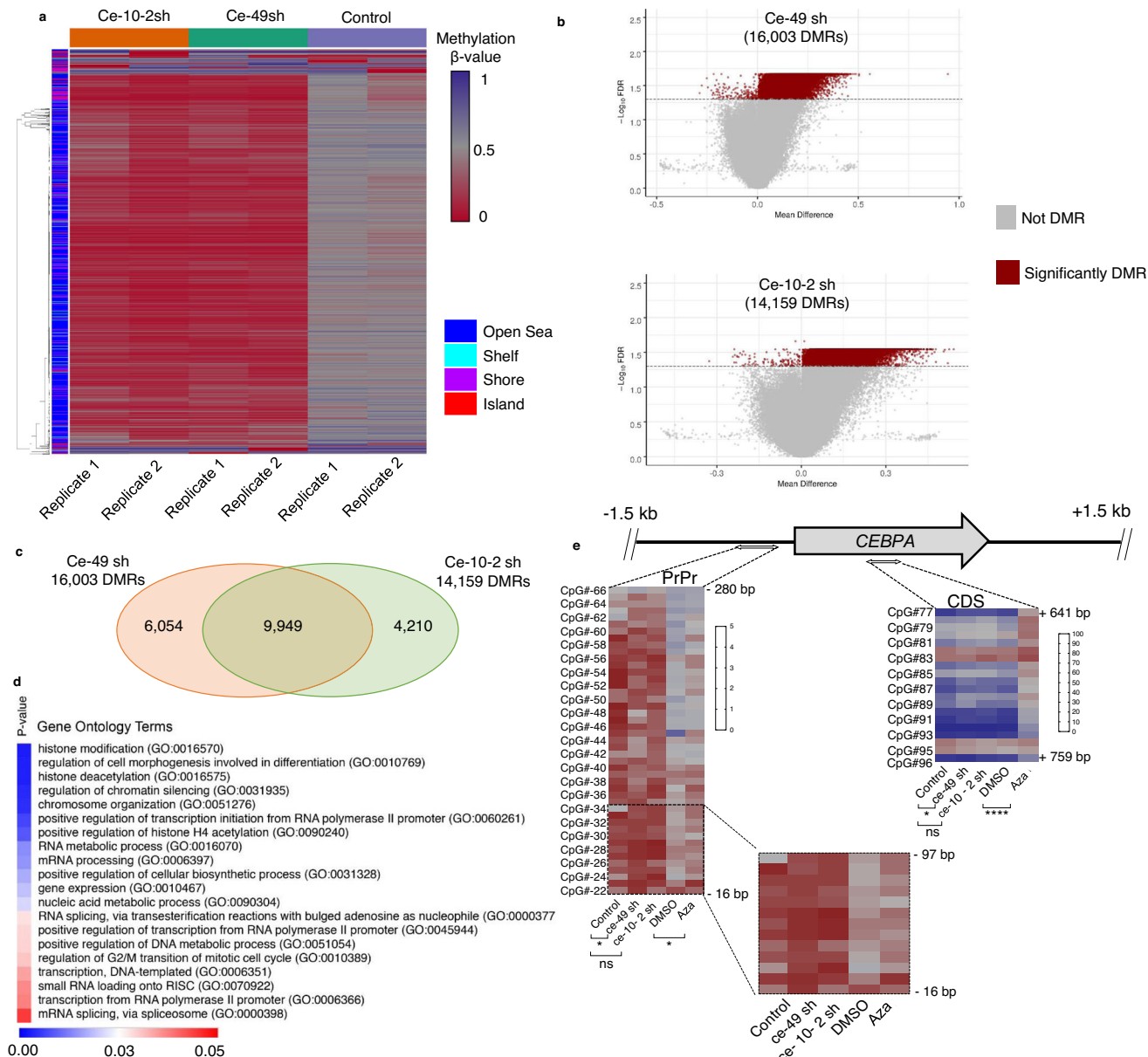

**Fig. 6 | DNA methylation analyses. a** Heatmap of differentially methylated CpG regions (DMR) in K562 transfected with Ce-49 sh, Ce-10-2 sh or control (Cont.) aptamers. **b** Volcano plots reporting the significantly DMRs for Ce-49 sh (*upper panel*) and Ce-10-2 sh (*lower panel*). **c** DMRs overlapping between Ce-49 sh and Ce-10-2 sh. **d** Heatmap of top ranked Gene ontology terms for genes corresponding to the overlapping hypomethylated CpGs shared by Ce-49 sh and Ce-10-2 sh. *P* values were calculated by Enrichr using Fisher's exact tests. **e** Heatmaps of *CEBPA* differentially methylated CpG regions in K562 transfected with Ce-49 sh, or Cont. and treated with DMSO or 5-Aza. The mean of two biologically independent samples is reported. Source data are provided as a Source Data file.

DNMT1, while Ce-10-2 sh shows a higher variability (Supplementary Fig. 19).

Overall, these findings suggest that Ce-10-2 sh–DNMT1 complex undergoes to greater rearrangement in the time scale used, revealing important structural determinants related to the aptamer sequence and responsible for the stability of the aptamer complexes with the target. The in silico differences between Ce-49 sh and Ce-10-2 sh interacting profiles support the observation of a different efficacy in DNMT1 targeting mediated by the two aptaDiRs.

The in silico results along with the functional characterization of both aptaDiRs suggested a stronger selectivity and stability for the Ce-49 sh – DNMT1 complex, thereby enabling us to carry out the additional downstream in vitro and in vivo analyses on this aptaDiR.

Collectively, the results validate the complex stability and confirm a similar binding modality between the selected aptaDiRs and the parental sequence with DNMT1, particularly for Ce-49 sh.

## aptaDiR Ce-49 sh induces gene expression and demethylation in vitro

To gain deeper insights into the functional effect of Ce-49 sh and assess whether the changes of DNA methylation, detected by the EPIC array, correlated to the respective gene expression levels, we analyzed the expression of a set of six genes linked to cancer progression, which were also highly demethylated in Ce-49 sh treated samples. As shown (Fig. 8a), the expression of all tested genes (*ZCCHC3, PAX5, PLOD2, GAD2, RABA9* and *FOXP1)* was efficiently upregulated two to five-fold upon Ce-49 sh transfection as compared to the control RNA.

Further, we demonstrated that the Ce-49 sh effect is reversible. Indeed, DNMT1 activity in K562 nuclear extract is fully recovered six days upon transfection and accordingly, *CEBPA* expression returned to the baseline levels (Fig. 8b).

In order to validate the therapeutic potential of the Ce-49 sh aptaDiR, we analyzed its function on cancer cell lines with different

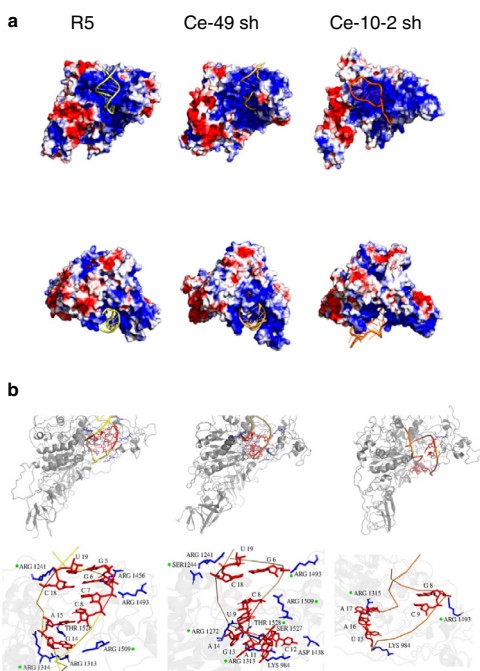

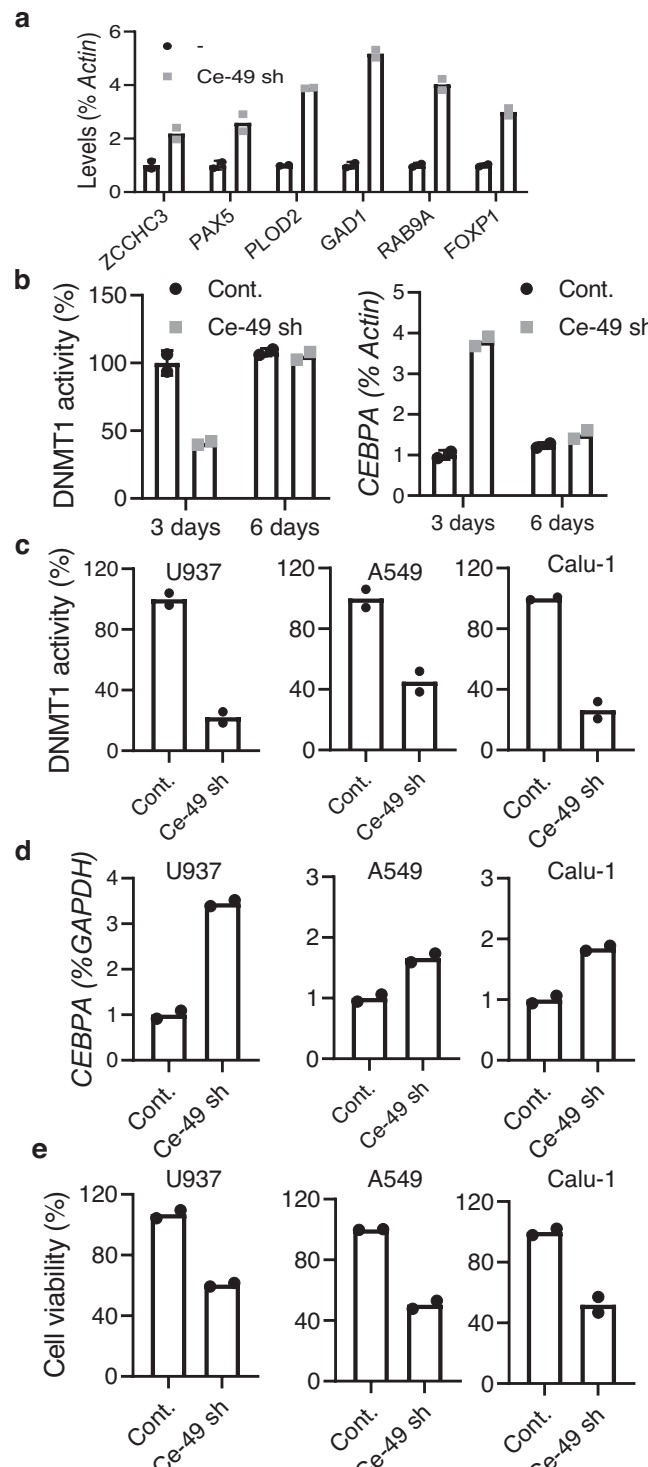

**Fig. 7 | Molecular dynamic analyses. a** Molecular dynamic representative structures of R5 or Ce-49 sh and Ce-10-2 sh aptamer-DNMT1 complex simulations, in explicit waters for 300 nanoseconds (ns), derived using a RMSD-based clustering approach as described in the method section. DNMT1 protein is shown in surface and coloured according to the electrostatic potential. The red colour (negative potential) arises from an excess of negative charges near the surface and the blue colour (positive potential) occurs when the surface is positively charged (± 3 kT/e). The white regions correspond to fairly neutral potentials. For aptamer regions, the following colour scheme was adopted: R5: yellow, Ce-49 sh: brown and Ce-10-2 sh: orange. **b** Molecular dynamic representative structures of R5 of Ce-49 sh and Ce-10-2 sh aptamer-DNMT1 complexes derived using a RMSD-based clustering approach among the overall frames of all the last 150 ns of the three replica runs. DNMT1 protein is shown in transparent cartoon and coloured in grey. For aptamers, the following colour scheme was adopted: R5: yellow, Ce-49 sh: brown and Ce-10-2 sh: orange. The top panels show a global view of the complexes. The bottom panels show a zoom of the anchor points between the protein (in blue sticks) and the aptamers (in red sticks), presented at least 20% among the concatenated trajectory of OL15 run simulations. The green dots indicate protein residues shown to be involved in persistent hydrogen bonds also during the simulation with the Parmbmsc1 force field.

levels of DNMT1 protein including the myeloid leukaemia U937 cells and the non-small cell lung cancer Calu-1 and A549 cells characterized by low or intermediate DNMT1 levels, respectively, as compared to K562 that displays high DNMT1 levels (Supplementary Fig. 20). As expected, inhibition of DNMT1 by Ce-49 sh (Fig. 8c), upregulation of *CEBPA* (Fig. 8d) and hampered cells' viability (Fig. 8e) to a similar degree was observed in all tested cell lines, confirming the high affinity of the aptaDiR for the target.

These results validate Ce-49 sh as a promising demethylating agent with great therapeutic potential.

## aptaDiR Ce-49 sh inhibits progression of human leukaemia in vivo

To validate the therapeutic potential of Ce-49 sh, we tested its efficacy against subcutaneous established K562 leukaemia *xenograft*s in immunodeficient NSG mice. Tumour-bearing mice were injected intratumorally every day for ten days (0.3 mg/kg/ injection) using Ce-49 sh encapsulated into commercially formulation of GenVoy ionisable lipid nanoparticles (LNPs)[21], 5-Aza (1 mg/kg/injection) or saline solution as a benchmark control (Fig. 9a). Tumour volumes were monitored daily. As shown in Fig. 9b, aptaDiR

**Fig. 8 | In vitro validation of Ce-49 sh functionality. a** Levels of indicated genes were analyzed by RT-qPCR in K562 cells transfected Ce-49 sh or Cont. for 72 h. **b** *Left panel*, DNMT1 inhibitor screening assay was performed with nuclear extracts recovered from K562 cells following 3 or 6 days from transfection with Ce-49 sh or Cont. Results are expressed as percentage relative to Cont. *Right panel*, Levels of *CEBPA* were analyzed by RT-qPCR in K562 transfected with Ce-49 sh or Cont. for 3 or 6 days. **c**–**e** Indicated cancer cell lines transfected with Ce-49 sh or Cont for 72 h. **c** DNMT1 inhibitor screening assay was performed with nuclear extracts. **d** Levels of *CEBPA* were analyzed by RT-qPCR. **e** Cell viability was measured. In (**a**–**e**) mean values and the corresponding data points are reported. Experiments are representative of two independent biological replicates (*n* = 2). Source data are provided as a Source Data file.

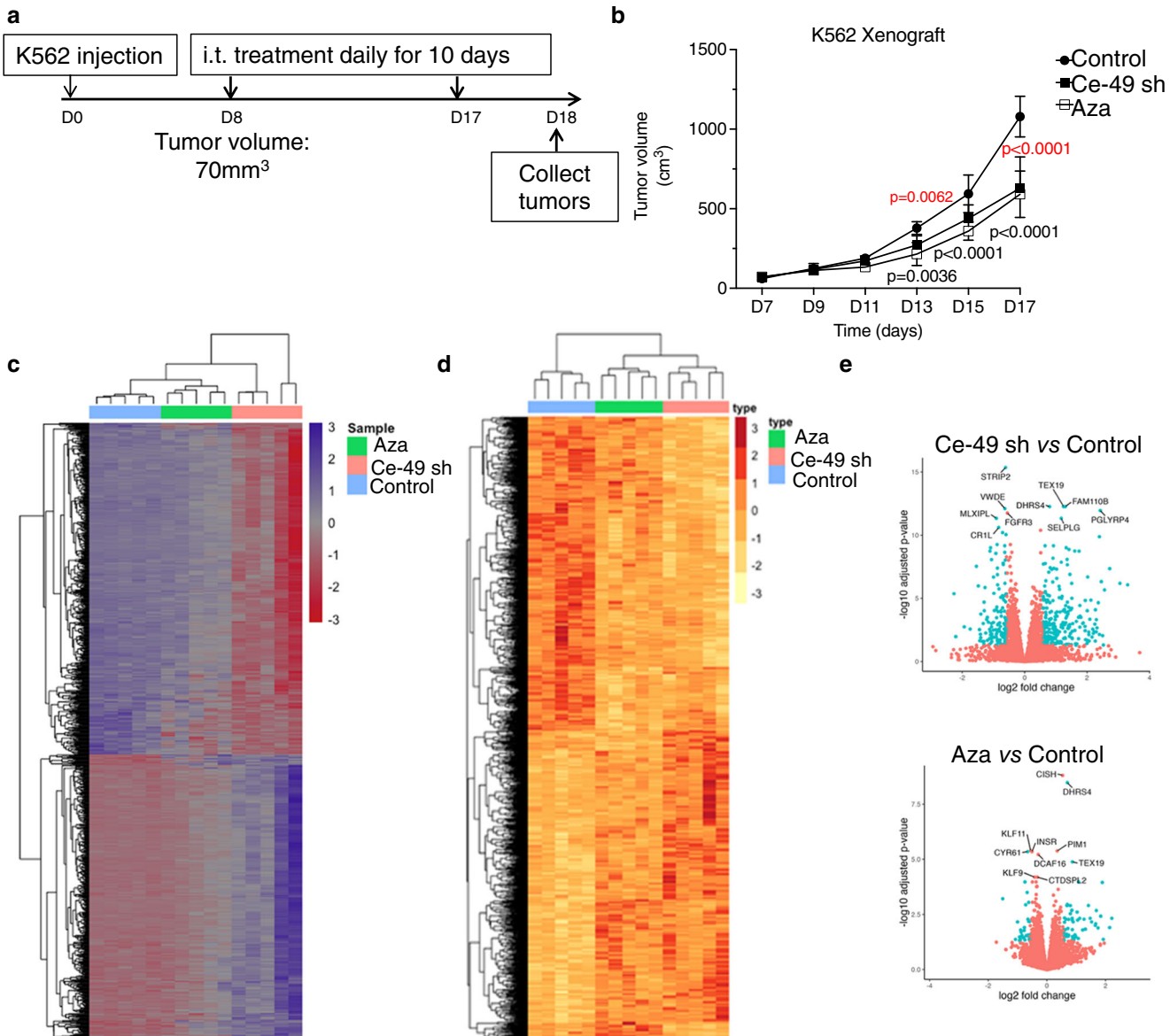

**Fig. 9 | In vivo validation of Ce-49 sh. a** Scheme of the in vivo experiment. (**b**–**e**) NSG mice bearing K562 tumours were daily-injected intratumorally with PBS (control) or Ce-49 LNPs or 5-Aza. Following 10 days of treatments, mice were then sacrificed. **b** Plot of tumour volumes measured by caliper. Error bars depict mean ± SD. Statistics of Ce-49 sh *versus* PBS (red) or 5-Aza *versus* PBS (black) by two-way ANOVA are reported (n = 5 biological samples). Source data are provided as a Source Data file. **c** Heatmap and cluster of the differentially methylated CpG sites (FDR < 0.05) in treated tumours. **d** Heatmap and cluster of differentially expressed genes (FDR < 0.05) in treated tumours. **e** Volcano plots of the differentially expressed genes. The significant differentially expressed genes are highlight in blue (FDR < 0.05, FC > 1.5). The top 10 genes (by FDR) are label with gene symbol.

treatment was as effective as 5-Aza in inhibiting tumour growth. Ten days after treatments, mice were euthanized to harvest tumours for DNA and RNA isolation and changes in the DNA methylation and gene expression levels measured. Methylome profile was assessed by the EPIC array platform. The differential methylation analyses (Fig. 9c) revealed significant reduction of DNA methylation in the aptaDiR and 5-Aza-treated tumours as compared to negative control (Fig. 9c). Significant gene reactivation was induced upon Ce-49 sh and 5- Aza treatment vs. the negative control (Fig. 9d, e). 1780 and 562 differentially expressed genes (FDR < 0.05) were observed in the Ce-49 sh and 5-Aza treated samples, respectively, when compared to the control. Remarkably, aptaDiR treatment induced a stronger DNA demethylation and gene reactivation than 5-Aza with a partial overlap in patterns of gene expression between these two treatments (Fig. 9e and Supplementary Fig. 21a), underlining the

different mode of action of the aptaDiR as compared to 5-Aza. Further analysis confirmed strong correlation between gene methylation and expression status for aptaDiR treated samples (Supplementary Fig. 21b).

These results provide evidence underscoring the efficacy of aptaDiR approach compared to a standard demethylation agent and suggest potential therapeutic application of this RNA-based technology to epigenetic regulation and cancer therapy.

## Discussion

The study herein presented aims at neutralizing the major epigenetic player, DNMT1, using a newly developed RNA aptamer platform—the aptaDiR. On the assumption that RNAs are epigenetic modulators capable of inhibiting the activity of DNMT1[15,22,23], we examined the possibility to merge this novel RNA feature to the key features

aptamers retain as ligands: the strong affinity and high specificity to a designated target.

We adopted an innovative doped SELEX strategy to isolate high-affinity DNMT1-specific RNA aptamers (Fig. 1). Put simply, for the initial pool of selection we exploited the propensity of DNMT1 to bind stem-loop RNA[15]. We designed and mixed together three libraries containing each a small degenerated region in the context of a short DiR sequence (R5) (Fig. 1). This approach allowed the rapid evolution (only three rounds of selection) of ligands exhibiting stronger affinity and higher stability than the endogenous DNMT1–RNA bait (R5).

Among the potential candidates, we shortlisted the most selective (with a $K_D$ of ~ 70 nM) aptaDiRs (Figs. 2 and 3). The aptaDiRs are unable to bind and inhibit the other two major DNMT family members: DNMT3A/3B, nor to bind to the unrelated proteins KAT5 and HSA (Fig. 4). These molecules, stably form in silico and in vitro complexes with DNMT1 and inhibit DNMT1 activity reducing cell viability as expected for DNMT inhibitors (Figs. 5–8). Most importantly, the selected lead aptaDiR candidate proved to be comparably as effective as 5-Aza in inhibiting tumour growth with even stronger DNA demethylation and gene reactivation effects in vivo (Fig. 9).

Currently approved hypomethylating agents, such as the cytosine nucleoside analogues, lack selectivity and are characterized by low stability and high toxicity[24]. Considering this, the herein aptaDiRs represent an alternative strategy to overcome the clinical limitations of the prevailing administered DNMT inhibitors.

Our platform includes the dissection of the structural and molecular factors that drive the aptaDiR–DNMT1 complex formation from an atomistic point of view, as determined by means of theoretical approaches. Even though we confirm that the stem-like-secondary structure of the aptaDiRs is needed for the DNMT1 binding, our analyses also suggest a potential role of sequence determinants in DNMT1–RNA interaction. The aptaDiR–DNMT1 in silico modelling reconciled the RNA stem-loop structural requirements, prerequisite for the DNMT1 binding, with the role of aptaDiR sequences and the biochemical analyses carried out for the aptaDiR validation. Further analyses will be needed to further confirm the role of the sequence determinants. Molecular dynamic simulations were performed using two force fields to limit the impact of the accuracy of the theoretical RNA descriptors[25–27]. Consistently, we identify the main hot spots of the complex formation. In summary, the higher binding of the R5 and Ce-49 sh if compared to Ce-10-2 sh derives from an extended network of interactions that promotes a particularly stable complex formation.

Recently, Wang et al. (2019)[28] have reported the generation of aptamers targeting DNMT1 using as a starting pool a high complexity DNA library. However, it remains counterintuitive to opt for a single stranded (ss) DNA aptamer when DNMT1 was shown to have a stronger affinity to RNA than DNA[15]. The aptamer described by Wang et al. (Apt. #9) displayed an affinity for DNMT1 of 0.77 μM, a value considerably higher than those detected for all the RNA sequences highlighted in our current and previous study[15]. The aptaDiRs herein reported have affinity values in the nanomolar range, thus 10 times lower than the one indicated for the ssDNA aptamer by Wang et al. Importantly, the stability of the RNA molecules, which display an half-life longer than 48 h in 85% human serum, considerably exceeds that of the reported Apt #9 with an half-life of 11 h in 10% serum media[28]. These differences lie in: (1) the selection strategy applied that preserves the secondary structure constraints required for DNMT1–RNA interaction while evolving aptamers with enhanced binding and stability for clinical translational purposes; (2) the RNA molecules which are more flexible and easier to manipulate to achieve stability as compared to ssDNAs[11]. Lastly, the approach presented by Wang et al., targets DNMT1 using ssDNA molecules that lack competitive advantage in two major instances. First, as DNA molecules themselves, ssDNA aptamers are unable to outcompete DNA for the binding to DNMT1, as previously shown[15]. Second, the ssDNA aptamers display extremely low binding and poor in vitro stability that rule out future translational applications. Finally, a careful assessment of DNA methylation and gene expression analyses upon the ssDNA aptamers delivery, that are imperative validation steps to assess the proposed protocol, are missing[28].

Recently, a reversible DNMT1- specific inhibitor with improved tolerability has been described[29]. Although promising, the compound functions by intercalating within the DNA-DNMT1 complex, thus acting as maintenance inhibitor, and therefore requiring a higher working concentration than a molecule directly inhibiting the enzyme. aptaDiRs, instead, block DNMT1 without intermediate molecules, and act at a considerably lower concentration than the one reported for the small molecules[29]. Our study did not investigate the aptaDiR specificity with respect to the well-described DNMT1 de novo and maintenance DNA methyltransferase activity[30]. However, being the RNA an endogenous inhibitor through its direct binding to the catalytic domain of DNMT1[15], the inhibition likely occurs without leaning toward one specific methyltransferase activity.

Collectively, aptaDiRs represent a great advance in the field of targeted demethylation as compared to other proposed approaches.

In summary, our study presents two distinct novelties. First, it describes an original selection strategy that translates the DNMT1-neutralizing properties of specialized endogenous RNAs (DiRs) into an aptamer-based platform leading to the aptaDiRs. In contrast with standard SELEX procedures, we designed a "doping" approach to introduce stringent constraints on the aptamer folding, a strategy well reported by other studies[31–33]. In such approach, a "total randomization" is replaced by a "delicate variation" of a defined sequence, the DNMT1 bait that was previously characterized for both sequence and structural constraints[15]. The selection resulted in generation of RNA aptamers with the same predicted conformation of the DNMT1 bait but enhanced binding properties, over 10 times stronger than the parental sequence. Remarkably, this strategy paves the way to a broader range of application to generate aptamer–targeting proteins binding nucleic acid and specifically structured motifs present in larger transcripts as non-canonical RNAs.

Second, it provides an RNA-based approach that is highly selective, chemically stable and molecularly versatile to control DNA methylation for therapeutic purposes. Undoubtedly, the aptaDiRs are much more attractive for clinical applications than other nucleic-acid-based compounds lacking these essential features. Moreover, aptaDiR-mediated DNMT1 inhibition is reversible thereby enhancing the attractiveness of such molecules since methylation is an essential process for normal cell regulation. Also, aptamer-based molecules offer the possibility to design specific "antidote" oligonucleotides to revert or regulate their therapeutic activity[34]. Furthermore, multiple strategies for an active delivery into the nucleus are now available[35,36] and the ability of stem-loop structured aptamers to enter cells and localize within the nucleus have been demonstrated[37,38]. That is a critical element for the successful outcome of future in vivo applications and the translational feasibility of the aptaDiR platform.

Taken together these results provide a proof-of-concept for the generation of innovative, designated, and cost-effective RNA-based epigenetic therapy. This strategy is not purely restricted to DNMT1 but conveyable to other epigenetic factors thereby broadening the clinical applications of RNA aptamers to multiple types of cancers or diseases associated with epigenetic alterations.

## Methods

### Library preparation and in vitro transcription
Randomized sub-libraries were purchased from Genomics and PCR amplified by using 0.05 U/μl Fire Pol DNA polymerase (Microtech) in a

mix containing: 0.4 µM primers, 0.2 mM dNTPs. After 3 min initial denaturation at 95 °C, the protocol used was: 10 cycles of: 95 °C for 30 s, 64 °C for 1 min, 72 °C for 30 s; following final extension of 5 min at 72 °C. Primers used were: forward (with T7 RNA Polymerase promoter): 5′ TAATACGACTCACTATAGGGCTGAAGGGGTTACTGGG-3′; reverse: 5′-CTCCTCCCCGGGGCAGATA-3′. Equal amounts of each sublibraries were mixed and transcribed. In vitro transcription was performed in the presence of 1 mM 2′-F-Py using a mutant form of T7 RNA polymerase (Y639F). DNA template was incubated at 37 °C overnight in a transcription mix containing: transcription buffer 1× (Epicentre Biotechnologies), 1 mM 2′F-Py (2′F-2′-dCTP and 2′F-2′-dUTP, TriLink Biotechnologies), 1 mM ATP, 1 mM GTP (Thermo scientific), 10 mM dithiothreitol (DTT) (Thermo scientific), 0.5 U/µl RNAse inhibitors (Roche), 5 µg/ml inorganic pyrophosphatase (Roche), and 1.5 U/µl of the mutant T7 RNA polymerase (T7 R&DNA polymerase, Epicentre Biotechnologies). After transcription, any leftover DNAs were removed by DNase I (Roche) digestion and RNAs were recovered by phenol:chloroform extraction, ethanol precipitation and gel purification on a denaturing 8% acrylamide/7 M Urea gel[39].

In vitro transcription in the presence of a mixture of 5′-biotin-G-monophosphate (TriLink Biotechnologies) and GTP (molar ratio 3:2) were performed for the biotinylation of the long aptamers.

### SELEX procedure
For the SELEX strategy, GST-tagged DNMT1 was used as target for the selection. Recombinant Human DNMT1 with an N-terminal GST tag was purchased from Active Motif and Pierce Glutathione Magnetic Beads (Thermo Scientific) were used to separate aptamer-GST-tagged protein complexes.

Before each cycle of SELEX, the 2′F-Py RNA pool was dissolved in RNAse free water and subjected to denaturation/renaturation steps of 85 °C for 5 min, ice for 2 min and 37 °C for 5 min. The RNA-protein incubation was performed in Binding Buffer (BB: 5 mM Tris-HCl pH 7,5; 5 mM MgCl$_2$, 1 mM DTT; 100 mM NaCl). At each cycle, the pool was first incubated for 30 min with Glutathione Magnetic Beads with a gentle rotation, as counter-selection step, and then the unbound RNA was recovered on a magnetic separator and used for selection. The recovered sequences were incubated with GST-tagged DNMT1 at room temperature for 30 min with a gentle rotation. Aptamer-protein complexes were purified on magnetic beads. The unbound was removed on a magnetic separator and the beads containing RNA-protein complexes were washed with BB. Bound RNAs were recovered by TriFast (Euroclone) extraction and RT-PCR and finally transcribed for the following round.

### RT-PCR
RNA recovered at each SELEX round was reverse transcribed by using M-MuLV reverse Transcriptase (Roche) in a mix containing a specific 5× buffer (50 mM Tris-HCl pH 8.3, 40 mM KCl, 6 mM MgCl$_2$, 10 mM DTT), 0.8 µM of reverse primer, 1 mM dNTPs. The protocol used for the reaction was: 30 min a t 42 °C and 30 min at 50 °C. The obtained product was then amplified by error prone PCR reaction in the presence of high MgCl$_2$ (7.5 mM) and dNTPs (1 mM).

### Cloning, sequencing and bioinformatics analyses
The final pool from SELEX was amplified by PCR, including in the programme a 15-min final extension at 72 °C to introduce A-overhangs. Individual sequences were cloned with TOPO-TA Cloning Kit (Invitrogen) according to the manufacturer's instruction. Single white clones were grown and DNA was extracted with plasmid Miniprep kit (Qiagen) and sequenced by Eurofins Genomics. Single aptamer sequences obtained were analyzed by using Multiple *Muscle* Algorithm. Aptamer secondary structures were predicted by using *RNAstructure* (https://rna.urmc.rochester.edu/RNAstructure.html).

### Synthetic oligonucleotides
Short RNAs were purchased from Trilink Biotechnologies or produced by the DNA/RNA Synthesis Laboratory, Beckman Research Institute of City of the Hope with 2′-F-Py all along the sequences.

Ce-9 sh: 5′ UGGGCUGAGCUCAUGGCGAGGCUUC 3′;
Ce-10-1 sh: 5′ AGGUAAUGGCGAGGCUUCUUAUCUG 3′;
Ce-10-2 sh: 5′ UUACUGGGCUGAGGUAAUGGCGAGG 3′;
Ce-49 sh: 5′ CUGAGGCCUAACGAAGGCUUCU 3′;
DNMT1 bait: 5′ CUGAGGCCUUGGCGAGGCUUCU 3′;
mutR5: 5′ CAGAGGACAAGGCGAGGCAACA 3′.

Biotin was added at the 3′-end of the sequences. Before each analysis, aptamers were kept 5 min at 85 °C, 2 min on ice and 5 min at 37 °C to enable the folding into their active conformations.

### ELONA assay
Microtiter High Binding plate (Nunc MaxiSorp) wells were coated with 30 nM of His-tagged DNMT1 (Active Motif) or HSA overnight at 4 °C. All subsequent steps were performed at room temperature. The plate wells were washed once with PBS and then blocked with 300 µL 3% BSA (AppliChem) in PBS for 2 h. After two washes, biotinylated aptamers dissolved in PBS (100 µL) were added for 2 h. After three washes, streptavidin-conjugated horseradish peroxidase (HRP) (Sigma-Aldrich) was incubated (1:10,000 dilution) for 1 h. Then, the plate was washed four times and the 3,3′,5,5′-Tetramethylbenzidine (TMB) substrate solution was added. The reaction was stopped with sulfuric acid (H$_2$SO$_4$) 0.16 M, forming a yellow reaction product. Signal intensity was measured with a microplate Reader (Thermo Scientific) at 450 nm.

### Human serum stability assay
Oligonucleotides were incubated at 4 µM concentration in 85% human serum from T0 to 7 days.

Type AB Human Serum provided by Sigma-Aldrich was used. At each time point, RNAs (4 µl, 16 pmoles) were recovered and incubated for 1 h at 37 °C with 2 µl of proteinase K solution (20 mg/ml) in order to remove serum proteins that interfere with RNA electrophoretic migration. Following proteinase K treatment, 12 µl of denaturing dye (95% formamide, 10 mM EDTA, Bromophenol Blue) was added to samples that were then stored at −80 °C. All time point samples were separated by electrophoresis into 15% acrylamide/7 M Urea gel. The gel was stained with ethidium bromide and visualized by UV exposure.

### Bio-Layer Interferometry technology system (BLI)
Bio-Layer Interferometry measurements were performed using a BLItz system and AR2G tips (Amine Reactive biosensors of 2nd Generation) (ForteBio Inc). After pre-hydration for 10 min in PBS buffer, the AR2G tips were efficiently functionalized with DNMT1 (Active Motif), DNMT3A (Abcam), DNMT3B (Abcam) and KAT5 (KAT5-1350H; Creative Biomart) proteins following the manufacturer's instructions. The AR2G tips enable the coupling of proteins to carboxylate groups on the biosensor surface via accessible amine groups. Accordingly, tips were activated with an EDC (0.2 M): NHS (0.05 M) coupling mixture for 300 s, then each ligand at the concentration of 20 µg/mL diluted in 100 mM MES pH 5.0 was exposed for 180 s to distinct biosensors. The average immobilization levels ranged between 2.0 and 2.5 nm. Unused activated carboxylated groups on the tips surface were reacted with 1.0 M ethanolamine hydrochloride, pH 8.5 for 120 s. After activation, a regeneration step with 10 mM NaOH was performed to minimize nonspecific binding. Blank biosensors were similarly prepared by activation and deactivation steps and were used to collect and subtract reference interferograms. For dose-dependent assays increasing concentrations of Ce-49 sh (concentrations ranging between 1.0 nM and 750 nM), Ce-10-2 sh (concentrations ranging between 10 nM and

750 nM) or DNMT1 bait (concentrations ranging between 100 nM and 2.0 μM) were used. Each run was performed following the reported steps: initial baseline (30 s), association (120 s) and dissociation (120 s), regeneration (2 × 30 s, 10 mM NaOH). The association step was performed using 4.0 μL of ligand solution placed in a drop holder drop, setting the shaker speed at 2000 rpm, according to the manufacturer's instructions. Duplicate or triplicate experiments were performed at increasing aptamer concentrations. Reference interferograms were subtracted from experimental values before data processing to reduce the background. Data were exported from the BLItz Pro 1.2 software and re-plotted with GraphPad Software. Plateau values of binding as reflected by changes in optical thickness (nm) at 120 s were used to calculate the affinity constant ($K_D$) by applying a non-linear curve fitting and one binding site hyperbola as model (GraphPad Prism).

### Microscale thermophoresis binding analysis

Binding measurements were carried out with a Monolith NT.115 device using MST Premium capillaries (Nanotemper Technologies GmbH). The MO Control software v1.5.3 from Nanotemper.

Technologies was used for data acquisition and manipulation.

The recombinant human His6-DNMT1 (rhDNMT1, Active motif, Catalog No: 31404) protein was labelled using the His-Tag labelling kit RED-tris-NTA 2nd Generation (Nanotemper Technologies GmbH)[40,41]. Labelling was performed according to the protocol provided by the manufacturer using rhDNMT1 at 100 nM in an optimized labelling buffer (25 mM Hepes pH 7.5, 250 mM NaCl and 0.05% Tween 20). The dye was dissolved at 100 nM in PBS-T buffer pH 7.4 according to the manufacturing instructions. The protein and the fluorescent dye solutions were mixed 1:1 obtaining a stock solution at 50 nM that was incubated for 30 min at room temperature in the dark. The sample was centrifuged for 10 min at 15,000 × g at 4 °C transferring the supernatant to a fresh tube. Aptamers Ce-49-sh, Ce-10-2 sh, DNMT1 bait and mutR5 used as a negative control were refolded and 1:1 serially diluted with binding buffer obtaining a total of 16 samples at final concentrations ranging from 1.5 μM to 35 pM in 10 μL total volume. To each sample, 10 μL of labelled DNMT1 solution was added and after 10 min at RT, samples were loaded into the MST Premium capillaries setting the instrument on 50% RED LED (650 nm excitation/670 nm emission) and medium MST power (40% MST in NT Control), operating at 25 °C. In each analysis, it was ensured that the intensity of the fluorescent target molecule was above 200 counts with variations <20% and that no protein aggregation occurred. Three independent technical replicates for each concentration were recorded and averaged. Measurements were analyzed using signals from MST-on times of 20 s for Ce-49 sh and 15 seconds for Ce-10-2 sh and DNMT1 bait, and with a signal-to-noise ratio >6 in the binding curves. According to the manufacturer's protocol, preliminary control experiments with a His6-labelled control peptide (provided by the manufacturer) and aptamers were performed to analyse possible interactions with the labelling dye. The lyophilized control peptide was suspended and further diluted in PBST-T to a final concentration of 200 nM. RED-tris-NTA was diluted in PBS-T at a final concentration of 100 nM. Then, 90 μL of the peptide solution was added to 90 μL of the dye (100 nM) and incubated for 30 min in the dark. Sixteen 1:1 serially diluted samples were prepared as described above. The final solutions of aptamers at concentrations ranging from 1.5 μM to 35 pM were analyzed in MST-runs.

To calculate the equilibrium binding constants, the MST traces of each capillary was analyzed for the difference between the baseline fluorescence, also referred to as Fcold, and the fluorescence level during the T jump or thermophoresis phases, known as Fhot. Both ΔFnorm or/and Fraction bound values were plotted against the ligand concentration in a concentration–response curve to provide an estimate of the affinity constants. Data were analyzed and fitted using MO.Affinity analysis software v2.2.7 (Nanotemper Technologies GmbH) by applying the $K_D$ model for binding interactions with a predicted 1:1 stoichiometry. Binding data were imported and analyzed with GraphPad Prism v4.00. The dissociation constant, $K_D$, was obtained by fitting a non-linear dose–response curve to a plot of Fnorm vs ligand concentration as detailed in Supplementary methods. Fnorm relates to the fluorescence values prior (F0) to and after (F1) IR laser activation.

### RNA electromobility shift assay

Aptamers (15 pmol) and DNA double-stranded oligonucleotides were end-labelled with [γ-$^{32}$P] ATP (Perkin Elmer) and T4 polynucleotide kinase (New England Biolabs). Reactions were incubated at 37 °C for 1 h and then passed through G-25 spin columns (GE Healthcare) according to the manufacturer's instructions to remove unincorporated radioactivity. Labelled samples were gel purified on 10% polyacrylamide gels prepared with 0.5x TBE buffer. Binding reactions were carried out in 10-μl volumes in the following buffer: 5 mM Tris, pH 7.4, 5 mM MgCl2, 1 mM dithiothreitol (DTT), 3% (v/v) glycerol, 100 mM NaCl. 0.8 μM of purified DNMT1 protein (Active Motif) was incubated with 30,000 cpm of $^{32}$P-labelled aptamers and single or double stranded, RNA or DNAs, respectively. All reactions were assembled on ice and incubated for 30 min at room temperature. Samples were separated on 6% native polyacrylamide gels (0.53 Tris/Borate/EDTA (TBE); 4 °C; 2.5 hs at 190 V). Gels were fixed, dried, and exposed to X-ray film.

### Aptamer-mediated pulldown

K562 cells were lysed with 10 mM Tris-HCl pH 7.5 containing 200 nM NaCl, 5 mM ethylenediaminetetraacetate (EDTA), 0.1% Triton X-100 and protease inhibitors. Extracts (500 μg in 0.5 ml lysis buffer) were incubated for 30 min with 200 nM heat denatured biotinylated aptamers with rotation. Following three washings with PBS cells, aptamer-protein complexes were purified on streptavidin beads (Thermo Fisher Scientific) for 2 h. Beads were washed three times with PBS and bound proteins were recovered by adding Laemmli buffer and then analyzed by immunoblotting with anti-DNMT1 antibody (Active Motif).

### Modelling and molecular dynamics simulations

The three-dimensional structure of the human DNMT1 protein was created by homology modelling with MODELLER 9.22 programme using the mouse DNMT1 in complex with its DNA substrate solved structure as template, downloaded from the Protein Data Bank database (PDB: 4da4)[42]. Three-dimensional models of R5, mutR5, Ce-49 sh and Ce-10-2 sh RNA aptamers were obtained employing the MC-Fold-MC-Sym pipeline[43] and optimized with 100 ns of MD simulations in waters. Aptamer representative conformations were extracted from each MD trajectories by clustering method and subsequently used to build the relative complex with the human DNMT1 modelled protein. In details, the complexes were built by manually docking[44] each aptamers conformation into the catalytic region of the human DNMT1 structure, using the DNA substrate coordinates as solved in the 4da4 pdb, as guide to define the binding pose. The initial complexes of human DNMT1 with R5, Ce-49 sh and Ce-10-2 sh aptamers were then subjected to MD simulations. MD simulations were run using the Parmbsc1 for DNA[25] with GROMACS 5.0.5[45] all the systems were solvated in an octahedron box using the TIP3P water models[46] with a 1.1 nm distance to the border of the molecule, simulating standard biological conditions by considering a 150 mM KCl concentration and additional ions to neutralize. Electrostatic interactions were treated using the particle mesh Ewald method and Berendsen algorithm to control temperature and pressure[47], following the indications dictated by the ABC consortium (https://bisi.ibcp.fr/ABC/Protocol.html) and previous protocols[48–53]. In all the systems, waters were firstly relaxed by energy minimization and 10 ps of simulations at 300 K, restraining the protein and RNA atomic positions with a harmonic potential. Then, the systems were heated up gradually to 300 K in a six step phases starting

from 50 K, finally the simulations were run in NPT standard conditions for 300 ns without restraints. GROMACS[45], VMD[53] and Pymol packages[54] were used to analyse all the MD trajectories. Clustering analyses of the last half time of each MD simulation were performed to extract representative conformations using the gromos clustering method with the algorithm described by Dura et al.[55]. For each cluster, the structure exhibiting the lowest RMSD relative to all the other members of the cluster was selected as representative.

We finally improved the accuracy of our results adding new simulations performed with amberOL15, a most recent force field properly designed to describe RNA systems. Each complex has been subjected to three additional runs of MD simulation for 300 ns using the same conditions described above implementing in GROMACS the amberOL15 force field. AmberOL15 force field, that includes ff14SB[56] force field for proteins, ff99bsc0χOL3 parameters for RNA, OL15[27] parameters for DNA Joung-Cheatham parameters for monovalent ions (as in frcmod.ionsjc_tip3p in AMBER package) Allner et al.[57] parameters for Mg2+. To derive the global percentage of hydrogen bonds between each aptamer and DNMT1 and a representative structure of the relative simulations, all the frames of the last 150 ns of each complex simulation were merged into a whole trajectory. To compute the hydration and ionic levels at the aptamer-protein interfaces, the number of water oxygens and ions has been calculated if within 5 Angstrom from both, protein and aptamer residues. The additional runs have improved the reliability of our models and have revealed that the impact of the force field used on the presented results is neglectable.

### Cells, transfection and inhibitor screening assay
K562 cells (#CCL-243), U937 cells (#CRL-1593.2), A549 (#CRM-CCL-185) and Calu-1 (#HTB-54) NSCLC cells were obtained from ATCC and grown in RPMI medium supplemented with 10% FBS (Sigma). Transfections were performed using serum-free Opti-MEM and Lipofectamine 2000 reagent (Thermo Fisher Scientific) according to the manufacturer's protocol. Cells were transfected with 100 nM of RNAs previously subjected to denaturation/renaturation steps. DNMT inhibitor screening assay (Abcam) was performed according to manufacturing' instruction using purified proteins or cellular nuclear extracts.

### Quantitative RT-PCR (RT-qPCR)
To analyze gene mRNA level, 1 μg of total RNA was reverse transcribed with iScript cDNA Synthesis Kit and amplified by real-time quantitative PCR with IQ-SYBR Green supermix (Bio- Rad, Hercules, CA, USA). ΔΔCt method was used for relative mRNA quantization by applying the equation $2^{-\Delta\Delta Ct}$.

Primers used were: human *CEBPA*: Forward: 5'CCGCTCCTCC ACGCCTGTCCTTAG-3'; Reverse: 5'-GCCCCACAGCCAGATCTCTAGGT C-3'; *ZCCHC3:* Forward: 5'-TCCTCGTCCGCATCTGTTTC- 3'; Reverse: 5'-GCAGCTCGATCT CGCATTTG-3'; *PAX5*: Forward: 5'-TCCGCCAGAG GATAGTGGAA-3'; Reverse: CGGAGC CAGTGGACACTATG; *PLOD2*: Forward: 5'-CACCGACGACCTCACTCAG-3'; Reverse: 5'-CCTTGAC-CAAGGACCTTCACA-3'; *GAD1*: Forward: 5'-GGGAACTAGCGAGAACG AGG-3'; Reverse: 5'-AATCGAGGATGACCTGTGCG-3'; *RAB9A*: Forward: 5'-TGGCCGCGAGACACTCT-3'; Reverse: 5'-AAGGATAGTCGCCGTT GTCC-3'; FOXP1: Forward: 5'-TGCCCATTTCGTCAGCAGAT-3'; Reverse: 5'-TGTGGTTGGCTGTTGTCACT-3'; *GAPDH* (housekeeping control): Forward: 5'-CTTTGTCAAGCTCATTTCCTGG-3'; Reverse: 5'-TCTTCCTC TTGTGCTCTTGC-3'; *β-actin* (housekeeping control): Forward: 5'-CA AGAGATGGCCACGGCTGCT-3'; Reverse: 5'-TCCTTCTGCATCCTGTC GGCA-3'.

### Cell viability assay
Cells were seeded in 96-well plates ($3 \times 10^3$ cells/well) and were transfected with indicated sequences (100 nM). Following 72 h, cell viability was assessed by CellTiter 96 Proliferation Assay (Promega).

### EPIC methylation array analysis in vitro
Genomic DNAs from K562 cells transfected with aptamers or control (mutR5) were extracted with DNeasy Blood & Tissue Kit (Qiagen). Samples were analyzed by Diagenode EPIC methylation array after bisulfite conversion.

The EPIC methylation array was analyzed using the Bioconductor package RnBeads (v2.4.0)[58,59] using the hg19 annotations. Background normalization method was set to "enmix.oob" with "swan" used for normalization. When setting up the sva covariates, only the "aptamers" designation was used. Differential methylation analysis was then performed comparing the respective "control" and "treatment" replicates. The heatmap output from RnBeads was used with minor adjustments for publication. For volcano plots, EnhancedVolcano (v1.4.0)[60] was used. CpG sites with an adjusted FDR < 0.05 were set to dark red for plotting. CpG sites were annotated to their closest gene using the variable annotations provided with the array platform using a custom R script. For Gene Ontology analysis, hypomethylated CpG sites (with mean.diff ≥ 0 value, FDR < 0.05) were selected, their HGCN gene ids extracted and analyzed using Enrichr[45,46]. The results reported herein correspond to the resulting "GO Biological Process" results.

### Pyrosequencing analysis (*CEBPA* and LINE-1)
K562 cells were transfected with Ce-49 sh or control (mutR5) or treated with 1.6 μM 5-Azacytidine (Sigma-Aldrich) resuspended in DMSO. Cells were collected at 72 h and genomic DNAs were extracted with DNeasy Blood & Tissue Kit (Qiagen). Samples were analyzed by EpigenDx, Inc.

### Lipid Nanoparticles (LNP) Production
The LNP-aptaDiRs were prepared in the delivery unit at the HIRM Precision RNA Medicine Core. Briefly, an aqueous phase and an organic phase containing ethanol and fluorescent labelled lipid nanoparticles were combined using a microfluidic mixer (Precision Nanosystems, Vancouver, Canada). After preparation, LNP formulations were dialyzed into 1× phosphate buffered saline (PBS) (pH 7.4) for 2 h in 10 K MWCO Slide-A-Lyzer dialysis cassettes (Thermo Fisher Scientific, Waltham, MA, USA). The size and zeta potential of LNP formulations was characterized. aptaDiR encapsulation efficiency was evaluated by low range Quanti-iT RiboGreen RNA as previously described (precisionnanosystems.com).

### In vivo study
K562 cells ($5 \times 10^6$ cells/mouse) were injected subcutaneously into the flank of NOD.Cg-Prkdc scidIl2rgtm1Wjl/SzJ (NSG) mice (strain #005557). After 8 days, mice bearing tumours were divided into three groups of five, matched by age and sex: (1) control group received PBS (100 μL/injection); (2) the second group received intra-tumoural injections of Ce-49 sh loaded GenVoy LNPs (Precision Nanosystems) (0.3 mg/kg injection/mouse in 100 μL/injection); (3) the third group received intra-tumoural injections of 5-Aza (Sigma) (1 mg/kg injection/mouse in 100 μL/injection). Treatments were performed daily for 10 days. Tumour size was determined by callipers according to the formula: tumour volume (mm3) = = $L \times W \times H$, where $L$ is length, $W$ is width and $H$ is height.

Mice were maintained at standard temperature (20–23 °C) with 30–70% humidity and a 12 light/12 dark cycle. At the end of the treatments, the mice were sacrificed, tumours were recovered and lysed for RNA/DNA extraction. Maxwell RSC tissue DNA kit (Promega, AS1610) and Maxwell RSC simply RNA cells kit (Promega, AS1390) were used. Genomic DNAs samples were analyzed by Diagenode EPIC methylation array after bisulfite conversion. RNA samples were subjected to RNA sequencing. Study was performed within the approved IACUC protocol 10016 of City of Hope.

### RNA-seq experiment for different expression analysis

The RNA-seq sequence data was aligned to the reference sequence (hg19) using STAR (ver2.7.9)[61] with Gencode annotation database (v19)[62]. RNA-seq quantification was measured as raw count using featureCount feature from Subread package (ver2.0.3)[63]. The small RNA genes (length <200 bps) and genes with low coverage (max expression less than 0.1 rpkm) are excluded for downstream different expression analysis. The different expression analysis of RNA-seq were applied using DESeq2 package (ver 1.26.0)[64] with FDR < 0.05 cutoff for different expressed gene. The heatmap is generated using R pheatmap (ver 1.0.12)[65] on scaled expression data. The volcano plot is generated using R ggplot2 (ver 3.3.5)[66].

### Statistics

For the statistical evaluation, we applied student't-test for caparison between two groups, one-way ANOVA with multiple comparisons for functional assays and two-way ANOVA for in vivo analyses. GraphPad Prism Software was used. Values of $p \leq 0.05$ were considered statistically significant. For GO ontology, p-values were calculated by Enrichr using Fisher's exact tests. This was then combined to calculate a combined score as described[67] for identifying the enriched terms listed in Supplementary Data 1.

### Reporting summary

Further information on research design is available in the Nature Portfolio Reporting Summary linked to this article.

## Data availability

Data are available on the gene omnibus database under the accession ID number: GSE154471 for the in vitro data; GSE205655 for the in vivo data. Raw data are provided with this paper in the Source Data file. Source data are provided with this paper.

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

## Acknowledgements

This work was supported by the: DOD BMFRP Idea Development Award W81XWH-20-1-0518, National Institute Of Diabetes And Digestive And Kidney Diseases of the National Institutes of Health under Award Number R01DK136116, HIRM Pilot Award 2021, NCI R00 CA188595, Italian Association for Cancer Research (AIRC) Start-Up grant #2014-15347, Fondazione Cariplo # 2016-0476 and Giovanni Armenise-Harvard Foundation to A.D.R.; the Worldwide cancer research grant #22-0129 and Italian Ministry of Health, GR-2011-02352546 to C.L.E.; Italian Ministry of Economy and Finance to the CNR for the Project FaReBio di Qualità and the Asian Fund for Cancer Research to V.D.F.; the Italian Ministry for University and Research (MUR) for the project PRIN no. 2015783N45 and project NEON, no. ARS01_00769 to M.R.; the Regione Campania for the projects NANOCAN and SATIN to M.R.; the Italian Ministry of Health, for the project "Research Project on CAR T cells for haematological malignancies and solid tumours" to M.R. S.U. was supported by the National Institutes of Health under Award Number 5T32HL007917-22; M.B. was supported by a PhD Fellowship from the Italian Ministry of Education, Universities, and Research to the University of Eastern Piedmont. This project was also partly supported by the National Cancer Institute of the National Institutes of Health under award number R01CA213131 to MK and P30CA033572 (City of Hope). We thank Prof. Daniel Tenen for constructive discussion and insightful suggestions and Fortunato Moscato for technical assistance. CINECA Supercomputing (Class C: aptNAMD - HP10CDH724 and Class C: MDpapt - HP10CL0DAJ) is acknowledged for computational support and the BIDMC ncRNA Core Facility for the synthesis of GenVoy LNPs (https://www.bidmc.org/research/core-facilities/ncrna-core). We thank EpigenDx, Inc. for DNA methylation analyses by pyrosequencing.

## Author contributions

A.D.R., C.L.E., M.K. and V.D.F. conceived and designed the study and wrote the manuscript; A.D.R.; C.L.E.; I.A.; A.S.; M.L.I.; D.W.; R.L.; S.U.; G.G.; M.B.; S.C. performed experiments; P.S. synthesized the aptaDiRs; H.L. and M.A.B. performed bioinformatics and statistical analyses; M.K., M.R., A.K.E. provided intellectual supports and revised the manuscript. I.A. and A.S. equally contributed. H.L. and M.A.B. equally contributed.

## Competing interests

The authors declare no competing interests.

## Additional information

[1]Institute for Experimental Endocrinology and Oncology "Gaetano Salvatore" (IEOS), CNR, Naples 80100, Italy. [2]Molecular Horizon, Bettona (PG) 06084, Italy. [3]Institute of Biostructures and Bioimaging, CNR, Naples 80100, Italy. [4]The Integrative Genomics Core, Beckman Research Institute, City of Hope Medical Center, Duarte, CA 91010, USA. [5]Cancer Science Institute of Singapore, National University of Singapore, 117599 Singapore, Singapore. [6]Harvard Stem Cell Institute, Harvard Medical School, Boston, MA 02115, USA. [7]Department of Immuno-Oncology, Beckman Research Institute, City of Hope National Medical Center, Duarte, CA 91010, USA. [8]Harvard Medical School Initiative for RNA Medicine, Harvard Medical School, Boston, MA 02115, USA. [9]Department of Translational Medicine, University of Eastern Piedmont, Novara 28100, Italy. [10]Department of Health Sciences, University of Eastern Piedmont, Novara 28100, Italy. [11]Department of Cancer Biology and Molecular Medicine, Beckman Research Institute, City of Hope Medical Center, Duarte, CA 91010, USA. [12]Ambition srl, Naples 80100, Italy. [13]Institute of Genetic and Biomedical Research (IRGB), CNR, Milan 20090, Italy. [14]Cancer Research Institute, Beth Israel Deaconess Medical Center, Boston, 330 Brookline Avenue Boston, Boston, MA 02215, USA. [15]Present address: Institute of Biostructures and Bioimaging, CNR, Naples 80100, Italy. [16]These authors contributed equally: Ida Autiero, Annamaria Sandomenico. ✉e-mail: c.esposito@ieos.cnr.it; MKortylewski@coh.org; vittorio.de-franciscis@irgb.cnr.it; adirusci@bidmc.harvard.edu

