## [Peer Review File · Nature Communications]

Targeted systematic evolution of an RNA platform neutralizing DNMT1 function and controlling DNA methylationReviewers' comments:

Reviewer #1 (Remarks to the Author):

Comments on Esposito et al, Nature Comm 2020

In this work, the authors build on their findings that a small RNA hairpin functions as an aptamer inhibitor of DNMT1, thereby modulating its ability to methylate DNA. Additional randomized sequences are appended to the core aptamer sequence and then a selection is performed to enhance DNMT1 binding. It is an interesting approach, but the paper suffers from an overall lack of sophistication, rigor and significance.

1. The authors have already published a claim that a simple hairpin aptamer can inhibit DNMT1, so the basic concept is not novel
2. Standards in the field for interpreting RNA-protein affinity or molecular interfaces are such that one would ordinarily solve a crystal structure in order to make explicit arguments about specificity, but this paper relies almost entirely on molecular modeling and hypothetical structures. Figure 1 is pure conjecture as it is based on simple RNA models and so it does not provide any insights.
3. Figure 3 presents putative binding data on the selected aptamers relative to the starting RNA, but it is completely qualitative in nature and the binding experiments themselves are not properly designed. There are no rigorous binding curves or quantitative analyses of K_d values to provide a basis for comparison. The authors compare relative changes in absorption intensities in such a way that the data not interpretable.
4. While Figure 4 and 5 present some binding isotherms obtained through various methodologies, the authors were unable to extract rigorous parameters from the experiments, again resulting in a qualitative overall analysis that does not provide explicit insights into mechanisms or basis for specificity.

Reviewer #2 (Remarks to the Author):

As I am theoretician, I will comment only on the modeling part of this paper. The latter has several issues that need to be addressed before the paper can be accepted for publication:

-Protein/RNA complexes are notoriously difficult to simulate (Krepl et al, J Chem Theory Comput, 2015 11(3):1220-43). At times, even simulations based on X-ray structures can lead to wrong results. This is even more likely when models are used, as in this case. The authors need to comment on the reliability of their models and on the impact of the latter on their results.

-We are not told which force field is used for the biomolecules.

-Water/counterions at the RNA/protein interface may play a key role for the binding, yet this is hardly discussed.

-The arguments brought to validate the stability of the complex (number of H-bonds, RMSDs) are qualitative. Their applicability in this work should be discussed.

Minor point:

-The “representative structures” in the Caption of Fig. 6 should be defined.

Reviewer #3 (Remarks to the Author):

This is an interesting manuscript describing the generation of an RNA aptamer platform exemplified by the development of a DNMT1 specific aptamer. The authors present an exhaustive characterization of the biochemical characteristics, including specificity of the lead 2 aptamers, also demonstrating some cellular activity. Although the study is novel, a recent study from Wang et al (Nucleic Acids Res 2019 Dec 16;47(22):11527-11537 and cited in the current manuscript) has recently described the development and activity of a DNMT1 aptamer. While the technology is different as well pointed out but the authors the degree of novelty of the current study is thus reduced.

Specific Comments

1. While the results clearly demonstrate specific inhibition of DNMT1, it is unclear whether both methyltransferase activity or any other functional activity is inhibited. Additionally, defining reversibility of DNMT1 inhibition is important as methylation is an essential process in the correct regulation and development of different cells.
2. Why do they only do 3 rounds of selection? How do authors rate that they have a correct enrichment? The authors should show and explain the characteristics or why they decide to select Ce-9 and Ce-10 as their best aptamers.
3. The authors analyze the interaction of Ce-9 and Ce-10 only in the K-562 cell line, which shows high levels of DNMT1. What happens in lines with a lower level of DNMT1 protein? The authors should perform these analyzes on 2-3 cell lines with different levels of DNMT1.
4. The analysis of the interaction of Ce-9 and Ce-10 should be analyzed against a greater number of epigenetic genes, not only KAT5, in order to show with greater certainty their specificity against DNMT1.
5. To show the functional activity of the DNMT1 specific aptamers, the authors evaluate the expression of CEBPA in K562 cell line after treatment with the aptamers. The evaluation of CEBPA expression is an indirect approximation to show the de-methylation of its promoter. The authors should demonstrate the CEBPA promoter methylation level by pyrosequencing strategy. It would also be necessary to

perform an analysis of global DNA methylation, either by dot-blot or the study of LINE-1 methylation levels by pyrosequencing, and in turn including in this analysis treatments with 5-Aza or decitabine, in order to demonstrate the efficacy of aptamers to inhibit the methyltransferase activity of DNMT1.

6. How do the authors explain the difference in DMRs obtained between Ce-9sh and Ce10-2sh? Why are these differences when both aptamer have shown similar degree of DNMT1 inhibition? Treatment with 5-Aza or decitabine should be included as a control in this analysis. The authors should carry out a pyrosequencing validation study of the results obtained using the EPIC array and in turn study whether the changes detected at the level of DNA methylation are also observed at the expression level of these genes.

7. To demonstrate the real efficacy of aptamers against DNMT-1, this study would need to include an analysis at the in vivo level, to demonstrate the functionality of the aptamers by inhibiting DNMT-1 at the in vivo level, detect changes at the methylation level of the DNA and compare it in turn to the efficacy of 5-Aza or decitabine.

Reply to the Referees

REFERENCE: Esposito CL et al. “Targeted systematic evolution of an RNA platform neutralizing DNMT1 function and controlling DNA methylation” (NCOMMS-20-32775-T).

We are grateful to all referees for the effort to evaluate and improve the manuscript. We believe that the referee’s comments and respective additions/corrections significantly strengthen the study. Following is the point-by-point replies to Reviewers’ comments indicating the changes introduced in the revised version of the manuscript (highlighted in blue).

Reviewer #1 (Remarks to the Author):

Comments on Esposito et al, Nature Comm 2020

In this work, the authors build on their findings that a small RNA hairpin functions as an aptamer inhibitor of DNMT1, thereby modulating its ability to methylate DNA. Additional randomized sequences are appended to the core aptamer sequence and then a selection is performed to enhance DNMT1 binding. **It is an interesting approach**, but the paper suffers from an overall lack of sophistication, rigor and significance.

1. The authors have already published a claim that a simple hairpin aptamer can inhibit DNMT1, so the basic concept is not novel

Our previous study demonstrates that specialized RNA termed DNMT1-interacting RNAs (DiRs), control DNA methylation establishment in a gene-specific manner (Di Ruscio et al, *Nature*, 2013). In the study we identified a natural DiR stem loop sequence of 22 nucleotides (reported as R5, and in the present manuscript referred to as DNMT1 bait) binding to and inhibiting DNMT1 (Di Ruscio et al, *Nature*, 2013). However, we did not test R5 or its derivative molecules as a strategy to alter DNA methylation and neutralize DNMT1 enzymatic activity and no other studies have reported such findings in our knowledge.

Herein, we used a “doped” SELEX approach to single out RNA-based nuclease resistant aptamers able to interfere with DNMT1 enzymatic activity. As a general rule standard protein-SELEX protocols require between 8 to 10 rounds of SELEX, starting from a high complexity library (10^{14} to 10^{15} variable sequences). In the “doping” strategy a “*total randomization*” is replaced by a “*delicate variation*” of a defined sequence – the R5 in our case (henceforth DNMT1 bait), that was previously characterized for both sequence and structural constraints. By such approach, we were able to scale down the library complexity nearly to 10^{12} -folds. Specifically, in contrast with SELEX procedures previously reported, our protocol was designed to introduce stringent constraints on the possible folding of the selected aptamers. This type of stringency was obtained by mixing multiple starting libraries of 2’Fluoro pyrimidine (F-Py)-modified RNAs wherein short parts of the bait were fully randomized, while the remaining portion was not. The selection resulted in generation of RNA aptamers with the same predicted conformation of the DNMT1 bait but enhanced binding properties, over 10 times stronger than the parental DNMT1 bait sequence. Consequently, we were able to select the candidates with the strongest affinity and highest specificity for DNMT1. Remarkably, this strategy offer a novel approach to target any given RNA-binding protein with a well-defined RNA-interacting module. We have clarified each point of this section in the revised version (see page 4 lines 87-90, page 5 lines 109-114 and the discussion of the revised manuscript).

2. Standards in the field for interpreting RNA-protein affinity or molecular interfaces are such that one would ordinarily solve a crystal structure in order to make explicit arguments about specificity, but this paper relies almost entirely on molecular modeling and hypothetical structures.

Figure 1 is pure conjecture as it is based on simple RNA models and so it does not provide any insights.

We acknowledge the point raised by the reviewer. However, it is worth noting that crystallographic studies on RNA aptamer-protein complexes are not trivial and frequently fail owing to a number of reasons mostly connected to the flexibility and the charge of these bio-molecules. Indeed, the number of aptamer/protein complexes deposited in the Protein Data Bank is still very modest, considering the global and growing interest on RNA-protein interactions. Actually, search of 'aptamer' in the Protein Data Bank only returns 148 structures having also proteins (out of 374 aptamers, in March 2022).

The *in-silico* modeling undertaken in our analysis had the purpose to create theoretical models grounded on solid experimental evidence and deepen the atomic detail of the molecular recognition. In every instance, modeling of RNA and RNA aptamer three-dimensional structures and respective binding with DNMT1 were guided by previously acquired experimental data. Molecular dynamic simulations were performed to study the structure, stability and dynamic of the complexes, focusing on the role of RNA sequences in the DNMT1 targeting, at the atomistic levels. This type of approach is largely accepted in the field to study nucleic acids and has been proved successful in advancing the knowledge on nucleic acid structures and overcoming some important limitations of experimental methods, as reviewed by Jiri Sponer *et al.* (Chem Rev. 2018 Apr 25; 118(8): 4177–4338).

To improve the accuracy of our models, we further expanded the sampling of the molecular dynamic trajectories by replicating the simulations three times for each complex system and using the most recent force field, properly developed for describing RNA molecules. Results have been included in the revised version (**Esposito et al. Figure 7** and **Supplementary Figure 15 to 18**; text at page 11 lines 253-256, page 12 and page 13 lines 280-287 in the revised manuscript).

3. Figure 3 presents putative binding data on the selected aptamers relative to the starting RNA, but it is completely qualitative in nature and the binding experiments themselves are not properly designed. There are no rigorous binding curves or quantitative analyses of K_D values to provide a basis for comparison. The authors compare relative changes in absorption intensities in such a way that the data are not interpretable.

We apologize with the reviewer for the confusion. The primary aim of Figure 3 was to provide a qualitative assessment of the binding between DNMT1 and the full length of several aptamer candidates emerging from the SELEX process (**Esposito et al. Figure 3a of the original version**) or their respective truncated variants used for the downstream analyses (**Esposito et al. Figure 3b, c of the original version**). The quantitative determination of the binding was measured for the short (truncated) aptamers through Bio-Layer Interferometry analyses (**Esposito et al. Figures 4a and 4b; and Supplementary Figures 5a and 5b, original version**) and the resulting K_D values are reported in the main text of the manuscript (please see reply to point 4).

As per reviewer's suggestion, we moved the plots as reported in **Supplementary Figures 5a and 5b** of the original version to the main **Figure 3 a-c** of the revised version in order to streamline the overall presentation and avoid further confusion. In addition, we have further confirmed the binding affinity of both aptaDiRs Ce-49-sh and Ce10-2-sh by microscale thermophoresis (MST) analyses (**Fig. R1**, please see below). The new results have been included in **Esposito et al. Figure 3 d-e** and **Supplementary Figures 6-8** of the revised version and described in the main text (page 8 lines 171-179).

Figure R1. Affinity analyses by MST. MST dose–response curves for the interaction analysis of indicated aptaDiRs reported as Fraction Bound.

4. While Figure 4 and 5 present some binding isotherms obtained through various methodologies, the authors were unable to extract rigorous parameters from the experiments, again resulting in a qualitative overall analysis that does not provide explicit insights into mechanisms or basis for specificity.

Although we understand the Reviewer's point that the experiments shown herein do not provide any indication on the mechanism of binding, we must reiterate that elucidating the mechanism of binding was neither an object nor an argument to support the conclusions of the study. Further, we respectfully disagree with the reviewer's statement that "... experiments result in a qualitative overall analysis...". Quite oppositely, results are obtained through a rigorous quantitative analysis and demonstrated the specificity of the binding to DNMT1 of the two shortlisted aptamers (Ce-49 sh and Ce-10-2 sh). Indeed, the binding isotherms obtained through Bio-Layer Interferometry as reported in **Supplementary Figure 5** (revised version) do show a strong, dose-dependent and saturable interactions of Ce-49 sh and Ce-10-2 sh to immobilized DNMT1. These interactions are underpinned by K_D s ranging between 79 and 66 nM, as determined by curve fitting of the plots reported in **Figures 3 a-c** (revised version). We do not understand what the referee means by "rigorous". The method reported in the manuscript is widely used and generally accepted to measure the affinity between biomolecules (Li, Zhenet *et al.* RSC ADVANCES, 2020, 10: 8181-8189; Lou, Xinhui; Egli, Martin; Yang, Xianbin Current protocols in nucleic acid chemistry 2016, 67: 7.25.1-7.25.15; Iaccarino E, *et al.* Biochem J. 2020, 477: 1391-1407).

Additional validation that the method provides a quantitative measurement of the interaction, is documented by the nearly 10-times lower binding affinity detected between the parental sequence DNMT1 bait and the protein ($K_D \sim 600 \text{ nM}$). Consistent with this observation, a small portion of the DNMT1 bait is captured by the DNMT1 immobilized on the sensor chip as compared to the Ce-49 sh and Ce-10-2 sh.

However, following reviewer's comment, we moved curve fitting plots as reported in **Supplementary Figures 5a** and **5b** of the original version to the main **Figure 3 a-c** of the revised version in order to improve the overall presentation and avoid further confusion.

Further, in the revised version, we included an additional quantitative analysis to characterize the aptaDiR affinity, by determining the K_D s using a MST approach (please refer to **Figure R1**, **Esposito et al. Figures 3d-e** and **Supplementary Figures 6-8** in the revised manuscript).

Pertaining the specificity, we have shown that the unfolded 2'-F-Py-modified RNA (mutR5) is unable to interact with the enzyme (**Esposito et al. Supplementary Fig 5** and **8** of the revised version), demonstrating a strong element of structural recognition between DNMT1 and RNA.

Remarkably, both Ce-49 sh and Ce-10-2 sh are unable to recognize fully unrelated (KAT5) and partially related (DNMT3A and DNMT3B) proteins (**Esposito et al. Figures 4a and 4b**). As a result, no binding curves can be extrapolated in these experimental conditions, given the very low response and the absence of a dose-response effect.

We acknowledge that we failed to properly address these points in the original version of the manuscript and have revised the text accordingly (page 8 lines 171-179).

Reviewer #2 (Remarks to the Author)

As I am theoretician, I will comment only on the modeling part of this paper. The latter has several issues that need to be addressed before the paper can be accepted for publication:

-Protein/RNA complexes are notoriously difficult to simulate (Krepl et al, J Chem Theory Comput, 2015 11(3):1220-43). At times, even simulations based on X-ray structures can lead to wrong results. This is even more likely when models are used, as in this case. The authors need to comment on the reliability of their models and on the impact of the latter on their results.

We agree with the reviewer that is difficult to simulate RNA-protein complexes when using theoretical models as the starting systems for molecular dynamic simulations that might lead to inaccurate results. This aspect is widely discussed in literature by Krepl and collaborators (Krepl et al, J Chem Theory Comput, 2015 11(3):1220-43).

However, our modeling approach was driven by solid experimental evidence reported in the presented manuscript and in our previous study (Di Ruscio et al, Nature, 2013) to dissect the atomistic details of the molecular recognition. The predicted binding between the RNA-aptamer and DNMT1 protein aimed at testing in-silico the sequence and structural requirements enabling such interaction. The models obtained for the RNA aptamer -DNMT1 interaction and the respective molecular dynamics reconciled the RNA stem-loop structural requirements, prerequisite for the DNMT1 binding, with the aptaDiR sequences and the biochemical analyses carried out for the aptaDiR validation (i.e. ELONA, EMSA, MTS and BLITZ analysis).

In summary, the molecular dynamic simulations were purposely used to describe at the atomistic levels the RNA aptamer-protein complexes in its three-dimensional conformation, and to assess possible perturbation in targeting DNMT1 owing to different RNA sequences. The trajectory analyses indicated atomistic details of various anchor points brought about by variations of the RNA sequences when complexed with DNMT1 protein and interesting differences among the binding modes showed by the three aptamers have emerged from our analyses. To further confirm our observations and improve the sampling of our trajectories, three additional runs for each complex have been performed with most recent force field describing RNAs, OL15. The new simulations confirmed the slightly diverse behavior of the three aptamers in targeting the DNMT1 and all the analyses have been added to the new version of the manuscript (**Esposito et al. Figure 7 and Supplementary Figure 15 to 18**; revised text at page 11 lines 253-256, page 12, page 13 lines 280-287 and page 16 356-367), punctually indicating the atomistic details of the aptamer-protein complexation, as peer reviewer's suggestion.

-We are not told which force field is used for the biomolecules.

We apologize with the reviewer for the missing details, they have been added to the revised manuscript version. The molecular dynamic simulations described in our manuscript have been conducted using the parmbsc1 force field, assuming this force could be a good compromise for both protein and nucleic acid parametrization as available in GROMACS (<http://www.gromacs.org>) at the time of the calculations. Then, additional trajectories using the currently available ff-nucleic-OL15 force field, which includes the RNA Olomouc dihedral refinements, and released in GROMACS on 2019, have been run for assessing the reliability of our trajectory analyses. Analyses have been added to the new version of the manuscript (**Esposito et al. Figure 7**,

Supplementary Figure 15 to 18 and **Supplementary Table 3** in the revised manuscript). Further, text has been revised to clarify how the model construction was guided by the use of experimental data, elaborating on the reliability of the simulations and indicating the force field applied (revised text at page 11 lines 253-256, page 12 and page 13 lines 280-287, page 16 lines 356-367 and page 28 lines 632-644).

- Water/counterions at the RNA/protein interface may play a key role for the binding, yet this is hardly discussed.

We thank the referee for the suggestion. We have analysed the role of water and counterions at the RNA/protein interfaces in mediating, favoring or stabilizing nucleic acid-protein interactions. The manuscript now comprise the water and ionic analyses for all the simulations performed (**Esposito et al. Supplementary Figures 15** and **16**, text at page page 11 lines 253-256 and page 12 lines 257-262 in the revised version) However, only a marginal difference emerged in the hydration of the Ce-10-2 sh-DNMT1 complexes interface during the trajectory performed with the amber Parmbsc1 force field, this behavior has not been confirmed in the additional runs. The trajectories performed with the OL15 force field all reveal a comparable behavior of water and ions among the diverse complexes.

-The arguments brought to validate the stability of the complex (number of H-bonds, RMSDs) are qualitative. Their applicability in this work should be discussed.

We appreciate the reviewer's comment and we have discussed both qualitative and quantitative features of our analyses in the revised manuscript, as suggested. Specifically, we addressed the qualitative role of molecular dynamic simulations in providing the structural description of the interaction and emphasize the scope of the trajectory analyses in revealing RNA key residues within the individual aptaDiR (revised text at page 11 lines 253-256, page 12, page 13 lines 280-287 and page 16 356-367). Moreover, the experimental data have been further evidenced in the revised text as a quantitative measure of the RNA aptamer-DNMT1 interaction i.e. ELONA, EMSA MST and BLITz analysis).

Minor point:

-The "representative structures" in the Caption of Fig. 6 should be defined

We defined the "representative structures" in the caption of Figure (**Esposito et al. Figure 7** of the revised version) in the revised version.

Reviewer #3 (Remarks to the Author):

This is an **interesting manuscript** describing the generation of an RNA aptamer platform exemplified by the development of a DNMT1 specific aptamer. The authors present and exhaustive characterization of the biochemical characteristics, including specificity of the lead 2 aptamers, also demonstrating some cellular activity. Although the study is novel, a recent study from Wang et al (Nucleic Acids Res 2019 Dec 16;47(22):11527-11537 and cited in the current manuscript) has recently described the development and activity of a DNMT1 aptamer. While the technology is different as well pointed out but the authors the degree of novelty of the current study is thus reduced.

We understand the reviewer's comment and acknowledge the apparent similarity between our work, and the previous report by Wang et al. However, there are compelling differences between

the RNA-based molecules we have developed, and the ssDNA aptamers described by Wang et al. as explained in the following points:

1) Conceptually, our manuscript aimed at designing an RNA strategy to control DNA methylation and reactivate genes aberrantly methylated in disease. We grounded our hypothesis on published and validated evidence that RNA binds to DNMT1, with a higher affinity than DNA, therefore inhibiting DNMT1's enzymatic activity (Di Ruscio et al, *Nature*, 2013). We reasoned that RNA molecules with enhanced structural stability and honed sequence specificity, but still interacting with DNMT1, could pave the way for the first generation of selective RNA-based DNMT1 inhibitors. *This rationale is absent from the works of Wang et al.*

2) The approach presented by Wang et al., targets DNMT1 with ssDNA molecules that do not show competitive advantage in two major instances. First, as DNA molecules themselves the ssDNA aptamers lack a fundamental prerequisite - the ability to outcompete DNA for the binding to DNMT1 as previously shown (Di Ruscio et al, *Nature*, 2013, **Figure 3h**). Second, the ssDNA aptamers presented by Wang et al., display modest binding affinities (K_D 's in the μM range) and poor *in vitro* stability (~11 hours in 10% serum media) (Wang et al., **Figure 3** and **Figure S14**). Contrary to this, our aptaDiRs show a strong affinity (nM K_D 's, Esposito et al. **Figure 3a-d**) with high specificity and structural stability (over 72 hours in 85% serum, Esposito et al. **Figure 2d**).

3) Wang et al., opted to select ssDNA aptamers using only the catalytic domain of DNMT1 in the SELEX protocol, while our aptaDiR selection was carried out using the full-length protein. It is well known that partial protein structures may not arrange and bind to the ligand as full-length proteins, and so our approach of selection using full-length DNMT1 is more robust and biologically consistent.

4) Wang et al. started their selection injecting a ssDNA library. Contrary to this, we adopted a targeted, more subtle variation-based strategy using the known DNMT1 bait. That per se enabled the introduction of stringent constraints on the possible folding of the selected aptamers, resulting in generation of RNA aptamers with the same predicted conformation and enhanced binding properties as compared to the DNMT1 bait.

5) Finally, and most relevant contextually, Wang et al. neither carefully assessed DNA methylation changes nor confirmed reactivation of genes aberrantly methylated upon the delivery of the ssDNA aptamers. Such assessments are imperative in assessing the validity of the proposed protocol. Instead, our works present significant changes in methylation and gene expression (Esposito et al. **Figures 6, 8** and **9**, revised version) induced by the aptaDiRs, validating our technique, approach and bespoke platform.

We also acknowledge that our report did not sufficiently contrast our methodological approach and results to those presented by Wang et al further compounding the unfortunate, observed similarity. These key points have been fully addressed in the revised version (see revised text at page 17 lines 373-387, page 18 lines 401-407).

Specific Comments

1. While the results clearly demonstrate specific inhibition of DNMT1, it is unclear whether both methyltransferase activity or any other functional activity is inhibited.

The referee raises an intriguing question. Number of evidence suggests a dual function of DNMT 1 as *de novo* and maintenance DNA methyltransferase (systematically and carefully reviewed in Jeltsch A and Jurkowska R.Z.; Trends in Biochemical Sciences, Volume 39, Issue 7, 2014). We intentionally avoided to introduce this additional layer of complexity in our study, given the likelihood for both activities to be inhibited, as the *in vitro* and in cell culture analyses, herein presented point to. Indeed, the RNA binds a portion of the catalytic domain corresponding to the

TRD (target recognition domain, located between aa 1300-1550, in the conserved domain 8 and 9). This observation is relevant in light of the fact that DNMT1 enzymatic inhibition occurs when an inhibitor molecule binds to: (1) the catalytic domain or (2) the regulatory domain (Svedruzic, Z. M. *Curr Med Chem* 15, 92-106, 2008; Jeltsch, A. *Epigenetics* 1, 63-66, 2006) and explains some *in vitro* data demonstrating that RNA inhibits DNMT1 methylase activity (Bolden et al. *J Biol Chem* 259, 12437-12443, 1984; Glickman JF et al. *Biochem Biophys Res Commun.* 230 (2):280-4, 1997; Pradhan S unpublished).

In every instance tested, RNA acts as a natural inhibitor of the DNMT1 catalytic activity and in view of the low processivity of the enzyme (Jeltsch A. *Epigenetics*. 2006; Svedruzic ZM, Reich NO. *Biochemistry*. 2005 Nov 15;44(45):14977-88; Pradhan S, Bacolla A, Wells RD, Roberts RJ. *J Biol Chem*. 1999 Nov 12; 274(46):33002-10.), it is unlikely that the inhibition would be specific to one methyltransferase activity.

In addition, aptaDiR can block DNMT1 without intermediate molecules, suggesting that both DNMT1 activities can be blocked. This aspect has been discussed in the revised manuscript (page 17 lines 393-394 and page 18 lines 395-396).

Additionally, defining reversibility of DNMT1 inhibition is important as methylation is an essential process in the correct regulation and development of different cells.

We agree with the referee's suggestion and have tested the reversibility effect of DNMT1 inhibition by the aptaDiR. Our analyses revealed that DNMT1 activity is fully recovered 6 days upon treatment with the aptaDiR Ce-49 sh (**Figure R2**, **Esposito et al. Figure 8b** and text at page 13 lines 296-298 in the revised version) In addition, an advantage of aptamers is the possibility to design specific "antidote" oligonucleotides to revert or regulate their therapeutic activity (Powell Gray et al. *Aptamers as Reversible Sorting Ligands for Preparation of Cells in Their Native State* *Cell Chem Biol* 2020, 27:232-244.e7; Nimjee et al. *Translation and Clinical Development of Antithrombotic Aptamers*. *Nucleic Acid Ther* 2016, 26:147-55). This key aspect has been discussed in the revised version (revised text at page 18 lines 413-416).

Figure R2. Reversibility of aptaDiR DNMT1 inhibition. *Left panel*, DNMT1 inhibitor screening assay was performed with nuclear extracts recovered from K562 cells following 3 or 6 days from transfection with Ce-49 sh or Cont. Results are expressed as percentage relative to Cont.. Mean \pm SD is reported (n=2). *Right panel*, Levels of CEBPA were analysed by RT-qPCR in K562 transfected with Ce-49sh or Cont. for 3 or 6 days.

2. Why do they only do 3 rounds of selection? How do authors rate that they have a correct enrichment? The authors should show and explain the characteristics or why they decide to select Ce-9 and Ce-10 as their best aptamers.

We apologize for not making our explanation clear enough. In the present manuscript, we employed a "doped" SELEX approach to identify RNA-based nuclease resistant aptamers able to interfere with DNMT1 enzymatic activity. As a general rule standard protein-SELEX protocols require between 8 to 10 rounds of selection starting from a high complexity library (10^{14} to 10^{15} variable sequences). Disparate to this, we utilized a "doping" strategy wherein "total randomization" is replaced by a targeted and "delicate variation" of a defined sequence – the DNMT1 bait in our case that we had previously characterized for both sequence and structural constraints. Thus, in contrast with SELEX procedures previously reported, our protocol was designed to introduce stringent constraints on the possible folding of the selected aptamers. This type of stringency was obtained by mixing multiple starting libraries of 2'Fluoro Pyrimidine-modified RNAs wherein only a

short part of the DNMT1 bait sequence was fully randomized, while the remaining portion was not. Consequently, we were able to scale down library complexity by nearly 10^{12} -fold. In this instance, few cycles are predicted to be necessary for the enrichment (William H Thiel, et al. PLOSOne 2012; doi: 10.1371/journal.pone.0043836), and 3 rounds were sufficient to select the candidates with the strongest and highest affinity and specificity for DNMT1. Indeed, the enrichment rate observed, well supported the starting hypothesis that conformational determinants rather than specific sequences represent the major constraints for the specific binding. Ce-10-2 sh and Ce-49 sh were chosen for further downstream analyses as they displayed the best dose-dependent binding with DNMT1 (**Esposito et al. Figure 3**). On a critical note, we foresee a broad applicability of this approach to target any given RNA-binding proteins with a known RNA-interacting module.

The rationale behind the use of a doping protocol and the choice of the selected candidates have been clarified in the main text (revised version at page 5 lines 109-114 and 118-119, page 6 line 120 and page 18 lines 401-407).

3. The authors analyze the interaction of Ce-9 and Ce-10 only in the K-562 cell line, which shows high levels of DNMT1. What happens in lines with a lower level of DNMT1 protein? The authors should perform these analyzes on 2-3 cell lines with different levels of DNMT1.

We have performed a western blot analysis on four different cell lines to assess the levels of DNMT1 protein. This screening showed that A549, Calu-1 and U937 displayed lower DNMT1 levels as compared to K562 (**Figure R3, Esposito et al. Supplementary Fig.19** of the revised manuscript). Thus, to address the reviewer's comment, we evaluated the effect of the aptaDiR in these cells. We found reduced DNMT1 activity and cell viability in all tested cell lines. *CEBPA* expression levels increased upon Ce-49 sh transfection, as expected given their respective methylation status. Results have been included in the revised version (**Figure R4, Esposito et al. Figure 8** and text at page 13 lines 299-302 and page 14 lines 303-305 in the revised version).

4. The analysis of the interaction of Ce-9 and Ce-10 should be analyzed against a greater number of epigenetic genes, not only KAT5, in order to show with greater certainty their specificity against DNMT1.

We have already analyzed the potential interaction of the aptaDiRs with the cognate DNA methyltransferase family members: DNMT3A and B, in addition to the unrelated histone acetyltransferase KAT5 (**Esposito et al. Figures 4 a-b and c** of the revised version), demonstrating for all of them a binding affinity 1000-fold reduced compared to that of DNMT1. The absence of binding to proteins highly related to DNMT1 (i.e. DNMT3A and B) strongly indicate the high specificity of aptaDiRs. In the revised text, these differences have been addressed and the results adequately discussed (page 9, lines 190-194).

5. To show the functional activity of the DNMT1 specific aptamers, the authors evaluate the expression of CEBPA in K562 cell line after treatment with the aptamers. The evaluation of CEBPA expression is an indirect approximation to show the de-methylation of its promoter. The authors should demonstrate the CEBPA promoter methylation level by pyrosequencing strategy. It would also be necessary to perform an analysis of global DNA methylation, either by dot-blot or the study of LINE-1 methylation levels by pyrosequencing, and in turn including in this analysis treatments with 5-Aza or decitabine, in order to demonstrate the efficacy of aptamers to inhibit the methyltransferase activity of DNMT1.

We agree with the referee and have monitored the methylation levels of *CEBPA* locus and LINE-1, by pyrosequencing, as per reviewer's suggestion. 5-Aza control was included as well. We confirmed the ability of Ce-49 sh to induce significant demethylation at *CEBPA* proximal promoter (-97 bp to -16 bp from the TSS) and coding region (+ 641 bp to + 750 bp from the TSS). As expected, Ce-49 sh did not alter the methylation levels of LINE-1 in contrast to 5-Aza treatment, in both replicates analyzed (please refer to **Figure R5**, and **Esposito et al. Supplementary Figure 10** in the revised manuscript). Ce-10-2 sh, instead, displayed an opposite pattern between the two replicates, suggesting a distinct mode of interaction as compared to Ce-49 sh. Overall, these findings indicate that a safer therapeutic profile can be achieved by Ce-49 sh aptaDiR. The new results have been added to the revised version (**Esposito et al. Figure 6e** and **Supplementary figure 10** of the revised manuscript) and the text modified accordingly (page 10 lines 232-233 and page 11 lines 234-239).

Figure R5 CEBPA methylation analyses. Heatmap of *CEBPA* methylated CpG regions in K562 transfected with Ce-49 sh, or Cont. or treated with DMSO or 5-Aza.

6. How do the authors explain the difference in DMRs obtained between Ce-9sh and Ce10-2sh? Why are these differences when both aptamers have shown similar degree of DNMT1 inhibition? Treatment with 5-Aza or decitabine should be included as a control in this analysis. The authors should carry out a pyrosequencing validation study of the results obtained using the EPIC array and in turn study whether the changes detected at the level of DNA methylation are also observed at the expression level of these genes.

The differences in DMRs obtained upon Ce-49 sh and Ce-10-2 sh treatment can be attributed to the stronger stability of the Ce-49 sh–DNMT1 with respect to Ce-10-2 sh–DNMT1 complex as indicated by: **a)** the distinct interacting profiles emerged with the *in-silico* modelling and resulting in a different mode of action of the two aptaDiRs (**Esposito et al. Figure 7** of the revised manuscript) and **b)** the inconsistent pattern of LINE 1 demethylation observed upon Ce-10-2 sh treatment (**Esposito et al. Supplementary Figure 10** of the revised manuscript. These points have been elaborated and discussed in the revised version (on page 11 lines 253-256, page 12 and page 13 lines 280-284).

Following reviewer's comment, we also correlated the aptaDiR-mediated changes of DNA methylation to gene expression. To this purpose, the expression of a set of six genes, among the top demethylated by the EPIC array were analysed by qRT-PCR. Results (**Figure R6**, reported in the revised manuscript as **Esposito et al. Figure 8a** and revised text at page 13 lines 290-295) confirmed the efficient up-regulation of all tested genes.

Furthermore, the aptaDiR's effect on DNA methylation and transcriptional activation was measured *in vivo* (please, see the following comment), including the 5-Aza treatment control, as per reviewer's suggestion.

Figure R6 In vitro validation of EPIC array. Levels of indicated genes were analysed by RT-qPCR in K562 cells transfected Ce-49 sh or Cont. for 72 hours. Mean \pm SD is reported (n=2).

7. To demonstrate the real efficacy of aptamers against DNMT-1, this study would need to include an analysis at the *in vivo* level, to demonstrate the functionality of the aptamers by inhibiting DNMT-1 at the *in vivo* level, detect changes at the methylation level of the DNA and compare it in turn to the efficacy of 5-Aza or decitabine.

Although our study provides a *proof-of-concept* for the generation of RNA-based epigenetic therapy, we agree with the reviewer that *in vivo* validation would improve the overall significance of the study and strengthen the conclusions. To this end, we took advantage of a subcutaneous established K562 leukemia *xenografts* in immunodeficient NSG mice. Tumour-bearing mice were injected intratumorally every days for ten days (0.3 mg/kg/ injection) using Ce-49 sh encapsulated into commercially formulation of GenVoy ionisable lipid nanoparticles (LNPs)²⁰, 5-Aza (1 mg/kg/injection) or saline solution as a benchmark. The effect on tumor growth and changes in DNA methylation and gene expression levels were analysed. Notably, aptaDiR resulted as effective as 5-Aza in inhibiting tumor growth and able to induce a stronger DNA demethylation and gene reactivation (**Figure R7**).

These data have been included in **Figure 9** of the revised version and discussed in the text (, page 14 lines 309-325, page 15 lines 326-331 and page 16 lines 349-351).

Figure R7 *In vivo* validation of Ce-49 sh. (a) Scheme of the *in vivo* experiment. NSG mice bearing K562 tumours were daily-injected intratumourally with PBS (control) or Ce-49 LNPs or 5-Aza. Following 10 days of treatments, mice were sacrificed and tumours were recovered for DNA/RNA extraction. (b) Plot of tumour volumes measured by caliper. Error bars depict mean \pm SD (n = 5). (c) Heatmap of differentially methylated CpG regions in treated tumours. (d) Heatmap and cluster of differentially expressed genes (FDR<0.05) in treated tumours. (e) Volcano plots of the differentially expressed genes. The significant differentially expressed genes are highlighted in blue (FDR<0.05, FC>1.5). The top 10 genes (by FDR) are labeled with gene symbols.

REVIEWER COMMENTS

Reviewer #2 (Remarks to the Author):

My issues have been properly addressed.

Reviewer #3 (Remarks to the Author):

All the issues raised have been adequately addressed, including additional studies

Targeted systemic evolution of an RNA platform neutralizing DNMT1 function and controlling DNA methylation, Esposito et al.

Reviewer #4 (Remarks to the Author):

Comments on Reviewer #1 report

1.

Reviewer #1 doubts the novelty of the present study. I disagree, because of the extensive analyses of aptaDiRs function on DNMT1 enzymatic activity, gene methylation, cell viability and tumor growth. In concordance with the authors response, this was not shown before to my knowledge. However, the authors claim the novelty of their used SELEX approach as well (in their response to point 1 and in the discussion, line 377) and here I have to object that SELEX methods with doped libraries were done before (e.g. recently by our lab: Geraci et al. 2022, ACS Chemical Biology).

2.

Reviewer #1 criticize the in silico modeling of the RNA-protein interaction. Here, I comply with the authors response that crystallographic studies are laborious and not trivial. However, I'm not familiar with the in silico modeling of the RNA-protein interaction used in the study and can't comment on the accuracy of the data.

3.

Reviewer #1 disagree with the binding data done after SELEX. The authors hereupon rearranged the figures and it is concise in my opinion. ELONA is widely used to investigate the binding of enriched libraries and single clones after SELEX. However, I would be also interested in the performance of the enriched library after 3 rounds of selection (Fig. 2a), and, the performance of the unmodified DNMT1 bait in comparison to the 2'F-Pyr-modified version (Suppl. Fig. 1).

4.

Reviewer #1 disapproves with the quantitative binding analyses. To address this point, the authors confirmed the acquired binding curves and KD values (by Bio-Layer interferometry) with a second

method, MST. Both methods give quantitative data and are common and accepted to show aptamer-target interactions. I'm convinced that the aptamers bind with the measured KD values in vitro. But I have to add that I miss the MST data of the DNMT1 bait.

The specificity was clearly shown by the aptamers ability to discriminate between DNMT1 related and unrelated proteins.

In my opinion, points 1, 3 and 4 of Reviewer #1 were addressed by the authors with the above mentioned minor inconsistencies. Apart from that, Supplementary Table 2 is missing in the manuscript, a few typos are present (e.g. line 165, 229, 270; Supplementary line 8, DNMT1 instead of DNMT1) and Supplementary figures 11, 12 and 13 are mixed up in the results description. Still, I would recommend this manuscript for revision.

Reply to the Referees

REFERENCE: Esposito CL et al. “Targeted systematic evolution of an RNA platform neutralizing DNMT1 function and controlling DNA methylation” (NCOMMS-20-32775A-Z).

We are grateful to all referees for the effort to evaluate and improve the manuscript. Following is the point-by-point replies to Reviewers' comments indicating the changes introduced in the revised version of the manuscript (highlighted in blue).

Reviewer #2 (Remarks to the Author):

My issues have been properly addressed.

We are thankful to the reviewer for appreciating the work performed.

Reviewer #3 (Remarks to the Author):

All the issues raised have been adequately addressed, including additional studies

We are glad that the revision has addressed all the issues raised by the reviewer and that she/he acknowledges the additional studies.

Reviewer #4 (Remarks to the Author):

1. Reviewer #1 doubts the novelty of the present study. I disagree, because of the extensive analyses of aptaDiRs function on DNMT1 enzymatic activity, gene methylation, cell viability and tumor growth. In concordance with the authors response, this was not shown before to my knowledge. However, the authors claim the novelty of their used SELEX approach as well (in their response to point 1 and in the discussion, line 377) and here I have to object that SELEX methods with doped libraries were done before (e.g. recently by our lab: Geraci et al. 2022, ACS Chemical Biology).

The reviewer is correct, and we apologize for failing to accurately report other examples of doped-based approach. The text has been modified (line 382 and lines 406-408 of the revised version) and previously described doped SELEX approaches cited (Refs 31-33, revised version).

2. Reviewer #1 criticize the in silico modeling of the RNA-protein interaction. Here, I comply with the authors response that crystallographic studies are laborious and not trivial. However, I'm not familiar with the in silico modeling of the RNA-protein interaction used in the study and can't comment on the accuracy of the data.

We are pleased to see that the reviewer agrees with our approach.

3. Reviewer #1 disagree with the binding data done after SELEX. The authors hereupon rearranged the figures and it is concise in my opinion. ELONA is widely used to investigate the binding of enriched libraries and single clones after SELEX. However, I would be also interested in the performance of the enriched library after 3 rounds of selection (Fig. 2a), and, the performance of the unmodified DNMT1 bait in comparison to the 2'F-Pyr-modified version (Suppl. Fig. 1).

We agree with the referee and appreciate their valid suggestion to evaluate by ELONA assay the performance of the enriched library and the unmodified DNMT1 bait in comparison to the 2'F-Pyr-modified version. To this end, we have analyzed the binding to DNMT1 of the enriched library after 3 rounds of selection (**Figure R1** and **Supplementary Figure 1** of the revised manuscript).

The results proved the good performance of DNMT1 binding for the enriched library and demonstrated that 2'F-Pyrimidine modification does not alter the binding of RNA to DNMT1. We have included the new data and modified the text accordingly (lines 106-107 and lines 131-132).

Figure R1. Binding of the DNMT1 bait and the enriched library on DNMT1 purified protein. (a) Binding ability of the enriched library after 3 rounds of selection on DNMT1 purified protein detected by ELONA. (b) Binding ability of unmodified R5 sequence (R5) or R5 modified with 2'-FPy (DNMT1 bait) on DNMT1 purified protein was detected by ELONA. In (a, b) Statistics by t-test: *, p<0.05.

4. Reviewer #1 disapproves with the quantitative binding analyses. To address this point, the authors confirmed the acquired binding curves and KD values (by Bio-Layer interferometry) with a second method, MST. Both methods give quantitative data and are common and accepted to show aptamer-target interactions. I'm convinced that the aptamers bind with the measured KD values in vitro. But I have to add that I miss the MST data of the DNMT1 bait. The specificity was clearly shown by the aptamers ability to discriminate between DNMT1 related and unrelated proteins.

We agree with the reviewer and have addressed his/her comment by performing MST analyses for the DNMT1 bait. Consistently, the results confirmed the results observed using the Bio-Layer interferometry assay. These new data have been included in the revised **Figure 2f** (here, **Figure R2**) and **Supplementary Figure 8** and the text revised accordingly (lines 176, 179 and 180).

Figure R2. MST analyses of DNMT1 bait. MST dose–response curves for the interaction analysis of DNMT1 bait reported as Fraction Bound.

In my opinion, points 1, 3 and 4 of Reviewer #1 were addressed by the authors with the above mentioned minor inconsistencies. Apart from that, Supplementary Table 2 is missing in the manuscript, a few typos are present (e.g. line 165, 229, 270; Supplementary line 8, DNMT1 instead

Reply to Referees: Esposito CL et al. (NCOMMS-20-32775A-Z).

of DNMT1) and Supplementary figures 11, 12 and 13 are mixed up in the results description. Still, I would recommend this manuscript for revision.

First, we are grateful to the reviewer for acknowledging our effort to address points 1, 3 and 4 of Reviewer #1. Second, we sincerely apologize for the inconsistencies although "minor" observed across the text. We greatly appreciate the referee for bringing them to our attention. As suggested, we have corrected these mishaps (throughout the manuscript) along with the table and figures' citations: Supplementary Table 2 has been included, Supplementary figures 11-13, now Supplementary figures 12-14 are properly listed (lines 235, 253 and 258).

REVIEWERS' COMMENTS

Reviewer #4 (Remarks to the Author):

All the issues raised have been adequately addressed, thank you.

Reply to Referees: Esposito CL et al. (NCOMMS-20-32775B).

Reply to the Referees

REFERENCE: Esposito CL et al. "Targeted systematic evolution of an RNA platform neutralizing DNMT1 function and controlling DNA methylation" (NCOMMS-20-32775A-Z).

We are grateful to all referees for the effort to evaluate and improve the manuscript. Following is the reply to Reviewer 4 comment.

Reviewer #4 (Remarks to the Author):

All the issues raised have been adequately addressed, thank you.

We are thankful to the reviewer for appreciating the work performed.